# A network embedding approach to identify active modules in biological interaction networks

Claude Pasquier[1] , Vincent Guerlais[1], Denis Pallez[1] , Raphaël Rapetti-Mauss[2], Olivier Soriani[2]

The identification of condition-specific gene sets from transcriptomic experiments is important to reveal regulatory and signaling mechanisms associated with a given cellular response. Statistical methods of differential expression analysis, designed to assess individual gene variations, have trouble highlighting modules of small varying genes whose interaction is essential to characterize phenotypic changes. To identify these highly informative gene modules, several methods have been proposed in recent years, but they have many limitations that make them of little use to biologists. Here, we propose an efficient method for identifying these active modules that operates on a data embedding combining gene expressions and interaction data. Applications carried out on real datasets show that our method can identify new groups of genes of high interest corresponding to functions not revealed by traditional approaches. Software is available at https://github.com/claudepasquier/amine.

## Introduction

Current high-throughput technologies are now capable of reliably quantifying, at the scale of an entire organism, the molecular changes that arise in response to diseases or environmental disturbances. To identify the most relevant genes for the process under study, statistical methods are commonly used to assign numerical values to the genes which reflect the degree of variation observed. In most studies, the genes considered of most interest are the ones whose relative differences in expression, or fold changes, are the largest. Unfortunately, the raw fold change is unreliable because it does not take into account for the inherent uncertainty of gene expression measurements. To overcome this uncertainty, existing methods calculate a $P$-value to reflect the statistical significance of the variation.

Selecting the genes of interest on the basis of fold changes, $P$-values or a combination of both makes it possible to compile a list of genes whose expression varies most significantly. However, this procedure fails to identify genes whose combined action is essential in the process under study but whose individual scores are too low.

Though, as pinpointed by Rapaport et al (2007), "a small but coherent difference in the expression of all the genes in a pathway should be more significant than a larger difference occurring in unrelated genes." Arising from this observation, many methods have been proposed to analyze gene activity in the light of our knowledge about their molecular interactions. These subnetworks are named "context-dependent active subnetworks" (He et al, 2017), "functional modules" (Beisser et al, 2010), "maximal scoring subgraphs" (Dittrich et al, 2008) or "altered subnetworks" (Reyna et al, 2018). The underlying idea is to identify a pertinent module of genes by simultaneously taking into account two criteria: one based on the measurement of the genes' activities and the other one reflecting the proximity between the genes in the module. One of the challenges is to define an appropriate scoring strategy based on these two criteria.

Nguyen et al (2019) classified main computational methods for solving the active subnetwork identification problem in six categories: (i) greedy algorithms, (ii) random walk algorithms, (iii) diffusion emulation models, (iv) evolutionary algorithms, (v) maximal clique identification, and (vi) clustering based methods. The first two methods are simple and rapid but are highly dependent to the starting point of the algorithm that does not guarantee to reach global optima. Conversely, methods (iii) and (iv) are able to find global optima (in accordance with the scoring system used) or an approximation of it at the prize of a computational burden. Method (v) does not fully answer the initial issue as it is probably not true that each gene involved in a biological process interacts with all the others. Finally, method (vi) offers the advantage of being based on existing clustering algorithms, but they require the calculation of a distance (or similarity) metric between objects. On an attributed graph, this distance must combine topological distances (such as the number of edges separating two nodes) with the similarity of the values associated with the nodes, making it challenging to determine the appropriate metric. Moreover, most of the clustering algorithms require to determine a priori the number of clusters to build, which is challenging. The commonality between all these

---

[1]Laboratoire d'Informatique, Signaux et Systèmes de Sophia-Antipolis, I3S - UMR7271 - UNS CNRS, Les Algorithmes - bât. Euclide B, Sophia Antipolis, France  [2]iBV - Institut de Biologie Valrose, Université Nice Sophia Antipolis, Faculté des Sciences, Parc Valrose, Nice cedex 2, France

Correspondence: claude.pasquier@univ-cotedazur.fr

methods is that their effectiveness is very dependent on the network topology. Unfortunately, it is known that molecular interaction networks are noisy and incomplete (Kondratyeva et al, 2022). In recent years, network embedding (Cui et al, 2019) has proven to be a powerful network analysis approach by generating a very informative and compact vector representation for each vertex $v$ in the network. The approach was initially considered as part of dimensionality reduction techniques (reducing, for example, a $|v| \times |v|$ adjacency matrix into a $|v| \times m$ matrix, where $m \ll |v|$). This dimensionality reduction allows to reduce noise and map nodes in a vector space in which distances between nodes accurately reflect their proximity in the original network.

Recent advances on deep learning has led to a plethora of methods based on deep neural networks for learning graph representations, methods that are often inspired by the learning of word embedding (Mikolov et al, 2013). Works on word embedding can be seen as learning linear sequences (word sequences). It has been shown that the resulting compact vector representations are capable of capturing rich semantic information about natural language. Processing graph structures is much more complicated. A popular approach is to convert a complex graph structure with a rich topology into a set of linear structures and then use a word embedding method to calculate the vector representation of each node. One of the most representative techniques for network embedding is Node2vec (Grover & Leskovec, 2016).

To date, network embedding has been used in a variety of computational biology studies, including predicting gene–disease associations (Ata et al, 2018; Peng et al, 2019), identifying essential proteins (Zeng et al, 2019; Wang et al, 2021a), predicting drug-target interactions (An & Yu, 2021), protein–protein interactions (Nasiri et al, 2021), drug–disease interactions (Zhou et al, 2020), and other biomedical data science problems (Su et al, 2020). All these studies are based on embeddings learned from unweighted graphs. Although some research efforts suggest computing embeddings on edge-weighted graphs, such as Node2vec+ (Liu et al, 2023), to our knowledge, there is currently no network embedding method specifically designed for node-weighted graphs. Furthermore, network embedding has never been applied to active module identification.

As mentioned above, the identification of active modules requires the simultaneous consideration of two criteria. In existing methods, measurements of gene activity and its network proximity are either combined to form a single metric or optimized simultaneously using multiobjective algorithms (Correa et al, 2019). When working on embedded networks, the proximity facet is embedded in the vector space. It is then possible to focus on the detection of subspaces containing genes that have a high activity. Consequently, the identified modules may not necessarily be fully connected in the original graph structure, in contrast to other methods that operate on the graph. Our approach thus favors the proximity of the nodes in the reduced vector space but there is no constraint for connectedness.

Following this line, we propose AMINE (Active Module Identification through Network Embedding), a new and efficient method for active module detection based on Node2vec (Grover & Leskovec, 2016). Our method uses a greedy approach to build the clusters based on the similarity of the nodes' encoding vectors and a metric

that takes into account the activity of the contained nodes. We evaluated the behavior of AMINE on artificially generated datasets on which it is possible to accurately measure the performance of the algorithms. On sparse interaction networks, in a task consisting of finding 3 distinct gene modules, AMINE outperforms the MRF method (Robinson et al, 2017), which itself achieved better results than four other methods published between 2009 and 2015 using the exact same dataset. On dense, more realistic networks, AMINE can locate modules with a higher accuracy than other evaluated methods. Furthermore, the work is done fairly quickly (30 min for a network of 10,000 genes) and without any parameterization. We next evaluated the performance of AMINE in predicting known pathways/biological processes on a publicly available transcriptomic dataset comparing pancreatic ductal adenocarcinoma (PDAC) with low and high metastatic potency. Finally, we explored in vitro unexpected functions predicted by AMINE for *BLIMP1/PRDM1*, one of the most overexpressed genes in pro-metastatic cells. Altogether, these analyses show that AMINE allows to complement the results obtained with classical approaches by identifying relevant functional groups of genes, and unveil unexpected functions.

# Results

### Evaluation of AMINE on artificial data generated by Robinson et al (2017)

Many studies dealing with the identification of active modules have tested their methods on datasets generated by themselves and which are, at times, difficult to reproduce. Robinson et al (2017) are among the few to give access to all materials used to test the MRF method they proposed. These materials contain the graph itself, the $P$-values associated with the nodes, and the modules to be identified. It gives us the opportunity to apply our method on exactly the same data.

The simulated experiment used to evaluate the MRF method (Robinson et al, 2017) consists of a set of 1,000 scale-free graphs, each containing 1,000 vertices associated with values simulated from a standard uniform distribution. In this dataset, each graph contains three distinct modules to be identified (called "hit modules"), with each module containing 10 vertices. To simulate the fact that the vertices belonging to these modules represent differentially expressed genes, and are therefore associated with low $P$-values, these vertices are assigned simulated values from a truncated Gaussian distribution with mean 0 and SD equals to 0.05. Robinson et al (2017) compared their MRF method with NePhe (Wang et al, 2009), Knode (Cornish & Markowetz, 2014), and BioNet (Beisser et al, 2010) and reported that MRF gives the best performance in terms of recall. Because it is known that there are exactly 30 true hits in the dataset, the authors rate the different methods by considering only the proportion of true hits in each hit list of size 30.

We ran AMINE on these data to detect the three most significant modules. The median recall of amine is 0.79, whereas MRF has less than 0.7. For AMINE, 50% of the recall score range between 0.73 and 0.84, whereas the same range for MRF is 0.6 and 0.75, respectively (Fig S1A). The precision (number of true positives divided by the

number of nodes identified) and the F1 score (harmonic mean of precision and recall) were also plotted on the same figure. The median value of the F1 score is 0.76, whereas the minimum and maximum values are 0.31 and 0.94, respectively. The total number of genes in the three identified modules range from 18 to 46 with a median of 28 (Fig S1B). This means that AMINE is able to identify modules close to the ground truth (although slightly smaller) without the need to specify their size a priori.

### Validation of the method on artificial dense networks

It has already been shown that simulating a biological network is a very difficult task (Pavlopoulos et al, 2011). For many years, Erdos and Reyni's model, which considers a network as a set of nodes connected in pairs with equal probability, was the dominant model (Erdős & Rényi, 1960). However, numerous studies of real networks have shown that these networks can self-organize into a scale-free state. In a scale-free network, the degree distribution of nodes follows a power-law, meaning that there are a few highly connected nodes (hubs) and many nodes with a low number of connections. Barabási and Albert (1999) have proposed a mathematical model, known as the preferential attachment model, for generating scale-free random networks. The principle of the method is to start with a small number of vertices ($m_0$) and to add, at each time step, a new vertex with m edges connecting the new vertex to m different vertices already present. The probability $\Pi$ that a new vertex is connected to vertex i depends on the connectivity $k_i$ of that vertex, such that $\Pi(k_i) = k_i / \sum_j k_j$.

However, we have found that the generation of artificial networks using this model, as carried out in various studies (for example, the articles of Cornish and Markowetz [2014] and Robinson et al [2017]) is too far from a real interaction graph for the results to be extrapolated (see Fig S2 for an example of such sparse graph).

In 2000, Barabási and Albert proposed an extended version of their model (Albert & Barabási, 2000) which enables more realistic networks to be generated. The same principle applies, but at each time step, one of the following three operations is performed: (i) with probability p, m new links are added; (ii) with probability p, m links are rewired; (iii) with probability 1 - p − q, a new node and m links are added. Our experiments suggest that using 3 initial nodes with parameters p and q set to 0.09 and 0.70, respectively, allows to generate random networks with topologies relatively close to real interaction networks. Details on the network generation, the choice of parameters p and q, and a comparison with real biological networks are given in the Materials and Methods section. Fig S3 shows an example of a generated dense network.

Using the extended Barabási–Albert model (Albert & Barabási, 2000) parametrized as specified in the Materials and Methods section, we generated 1,000 artificial networks with topologies relatively close to real interaction networks and one hit module to discover with size of 10 or 20 nodes. The performance of AMINE was compared with the methods GiGA (Breitling et al, 2004), BioNet (Beisser et al, 2010), COSINE (Ma et al, 2011), DIAMOnD (Ghiassian et al, 2015), DOMINO (Levi et al, 2021), and a baseline consisting in simply picking the genes with the lowest *P*-values. The results are shown in Fig 1 for networks with a module of size 10 and in

Fig S4 for networks with a module of size 20, both comprising 1,000 vertices.

Our results indicate that identifying a module on a denser network is a much more complicated process, as the median F1 score drops significantly from 0.76 on a sparse graph to values just above 0.5 (Figs 1A and S4A). Overall, however, the scores for the other methods are lower. COSINE and DIAMOnD scores are lower than the baseline strategy that relies solely on *P*-values. The F1 score obtained by BioNet is slightly below the baseline for the identification of an active module of size 10 and slightly above for the task of identifying an active module of size 20. In the case of COSINE and BioNet, these poor results come from predicting large modules with a median size exceeding 150 nodes for COSINE and 75 nodes for BioNet (Figs 1B and S4B). GiGA is the second-best performing method. For modules of size 20, the F1 scores of GiGA and AMINE are very close (Fig S4A). However, it should be noted that GiGA uses a parameter that determines the maximum size of the module to be identified. In our experiment, we set the maximum size equal to the expected size of the module to be identified, which, of course, facilitates the procedure. As shown in Figs 1B and S4B, for GiGA, the size of the identified modules is always smaller than the maximum. The other method that needs the expected module size is DIAMOnD which produces a module with exactly the specified size. DOMINO, a method that does not need the expected module size as a parameter identifies on average, for both configurations, small modules of less than 10 genes. For AMINE, without any indication on the size of the modules searched, we can see that the method predicts modules with a median size close to the ground truth size.

The results obtained by the DOMINO algorithm are atypical, because it identifies modules that closely match the ground truth on some networks, whereas on others, it completely misses the modules that need to be identified, leading to poor results. We believe that the fundamental assumption on which the algorithm was designed is the cause of this issue. In their article, Levi et al (2021) noted that modules detected by active module identification methods often include GO terms that are also found in modules identified on randomly permuted data, indicating that they are not specific to the biological context of the omics dataset being analyzed. The authors address this bias by partitioning the network into disjoint, highly connected subnetworks, which they term as "slices." This process is carried out statically because it only deals with the network of interactions and does not take into account the values associated with the nodes. The search for active modules is then performed within each of these slices. This method therefore relies on the strong assumption that an active module must be included in one of the slices identified on the network, which may not be true on artificially generated networks where an active module may consist of a set of connected genes that are not necessarily clustered together. This hypothesis could well be verified on real interaction networks, but this remains to be proven. Meanwhile, as far as artificial datasets are concerned, this specificity of the DOMINO algorithm explains, in our opinion, to a large extent, the poor performance of the method.

### Scalability of the method

To test to what extent our method is able to scale to larger networks, we applied it to an artificial network of 10,000 vertices that

A)

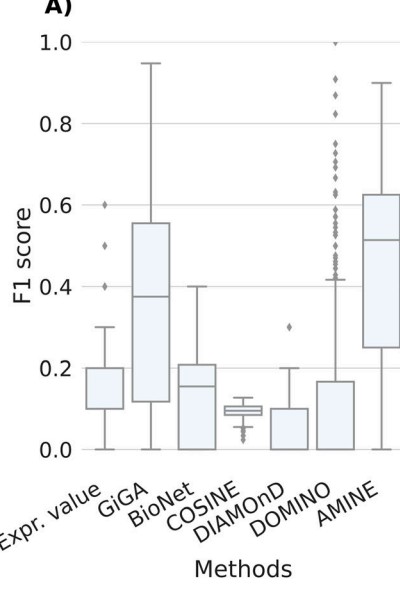

B)

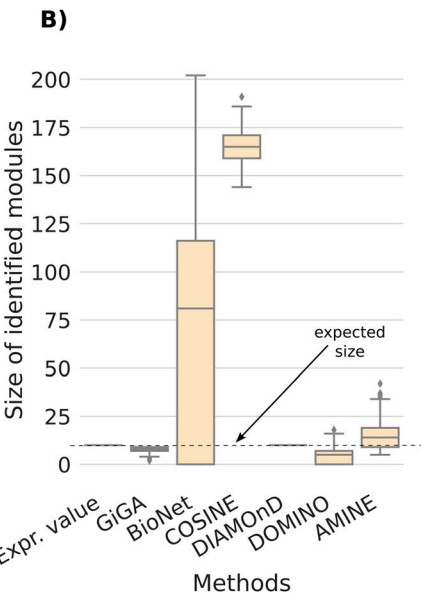

**Figure 1. Results obtained for the identification of modules of size 10.**
Comparison of AMINE with GiGA, BioNet, COSINE, DiAMOND, and DOMINO on a task consisting of identifying a module of size 10 on a dense artificial network with 1,000 vertices (on these networks, using the generation method described in the article, we count an average of 4,300 edges). The boxplots summarize the results obtained on 1,000 different networks. "Expr. value," that is used for baseline, consists of selecting the 10 genes with the lowest *P*-values. **(A)** Boxplots representing the distribution of F1 scores. **(B)** Boxplots representing the distribution of identified module sizes. The dotted line represents the expected size of the modules.

was generated using the same parameters previously defined. The processing time increases from 1 min for a dense network of 1,000 nodes to 30 min for a dense network of 10,000 nodes.

The distribution of F1 scores and the sizes of identified modules for AMINE and five other methods are displayed in Fig S5, and in Fig S6 for modules of sizes 10 and 20, respectively (The COSINE method has not been tested on 10,000-node networks because of its slow speed and poor performance). For modules of size 10, the median F1 is in agreement with that obtained on networks of size 1,000 (Figs S5A and S6A). The overall performance is however less good because it is penalized by the fact that, in many cases, the module is completely missed (which explains why the second quartile starts at zero in Fig S5A). This can be explained by the fact that, as the number of nodes in the network increases, the number of non-hit nodes that are associated with a random value higher than the values associated with hit genes increases. So, the probability that random modules score higher than the hit module increases too. This effect is less important for modules of size 20, for which the method works and for networks of 1,000 nodes (Fig S6A). Regarding sizes, they are still close to the ground truth, although there is a greater spread of values for modules of size 20 (Figs S5B and S6B). Except for GiGA, which performs reasonably well, all other methods display poor results.

**Validation using a real gene expression dataset**

To test the ability of AMINE and other methods to identify relevant biological functions, we downloaded from Gene Expression Omnibus a dataset relative to a study aimed at characterizing processes and genes associated to metastatic spreading in PDAC. With a 5-yr survival that has not significantly evolved for 30 yr despite progresses in anticancer therapies (<6%), PDAC is a cancer with one of the bleakest prognoses of the most fatal cancers. In the study of Chiou et al (2017), the authors compared two

populations of primary PDAC cells according to the expression of HMGA2, a gene associated to a high metastatic potency and poor outcome in several cancers, including PDAC. RNA-Seq quantification was carried out from six pairs of HMGA2+/HMGA2− cell populations, each pair originating from PDAC primary tumors spontaneously generated in a genetically engineered PDAC mouse model (PKC mice). Although the study identified hundreds of genes with consistent and significant differences between HMGA2- and HMGA2+ cells, the authors did not observe any significant signatures or enrichments associated with PDAC metastasis (gene set enrichment analysis and GO enrichments), excepted an overwhelming enrichment for hypoxia-induced genes in metastatic cells. Subsequent extensive experiments conducted in silico, in vitro, and in vivo demonstrated that BLIMP1, one of the top up-regulated genes in metastatic HMGA2+ cells, contributed to a subset of hypoxia-associated gene expression programs, leading to epithelial–mesenchymal transition (EMT), migration, and glucose metabolic reprogramming which are some of the hallmarks of PDAC metastatic status (Chiou et al, 2017; Wang et al, 2021b; Whittle et al, 2015). To assess the usefulness of the active module identification methods and enable cross-method comparisons, we used the considered algorithms to analyze the differential gene expression of the HMGA2+/HMGA2- RNA-Seq experiment performed using DESeq2 (Love et al, 2014). The goal was to determine whether it was possible to identify gene modules involved in the functions associated to the metastatic signature of cancer cells, for example, EMT, ECM reorganization, glycolytic reprogramming, angiogenesis, involvement of RAS and PI3K/ AKT signalling pathways. All the methods tested on the artificial datasets have been executed except COSINE which is a very slow method and always generates large modules of little interest. Parameters used to run the methods are specified in the Materials and Methods section. GiGA, BioNet, DIAMOnD, DOMINO, and AMINE have identified 5, 104, 1, 8, and 193 modules, respectively. Table S1 shows the top 5 modules identified by the different methods. We

**Table 1. Summary of the enrichment with GSEA of each identified module.**

| | | EMT | ECM | Glycolysis | Angiogenesis | RAS | PI3K | Hypoxia |
|---|---|---|---|---|---|---|---|---|
| GiGA | Module 1 | $7.78 \times 10^{-31}$ | $2.23 \times 10^{-6}$ | | $8.76 \times 10^{-6}$ | $2.16 \times 10^{-11}$ | $1.52 \times 10^{-12}$ | $2.19 \times 10^{-2}$ |
| | Module 2 | | | $1.99 \times 10^{-8}$ | | $3.4 \times 10^{-9}$ | | $5.67 \times 10^{-8}$ |
| | Module 3 | | | | | | | |
| | Module 4 | | $1.7 \times 10^{-2}$ | | | $1.27 \times 10^{-3}$ | | |
| | Module 5 | | $2.9 \times 10^{-2}$ | | | | | |
| BioNet | Module 1 | | | | | | | |
| | Module 2 | | | | | | | |
| | Module 3 | | | | | $1.8 \times 10^{-2}$ | | |
| | Module 4 | $6.33 \times 10^{-14}$ | $1.5 \times 10^{-9}$ | | | | $5.4 \times 10^{-11}$ | |
| | Module 5 | | | | | | | |
| DIAMOnD | Module 1 | $8.63 \times 10^{-25}$ | $9.38 \times 10^{-43}$ | | $1.07 \times 10^{-3}$ | | $6.97 \times 10^{-26}$ | |
| DOMINO | Module 1 | $3.10 \times 10^{-30}$ | $6.43 \times 10^{-27}$ | | $6.18 \times 10^{-6}$ | $6.96 \times 10^{-8}$ | $2.59 \times 10^{-16}$ | |
| | Module 2 | | | $5.28 \times 10^{-10}$ | | | | $6.44 \times 10^{-9}$ |
| | Module 3 | $9.88 \times 10^{-6}$ | $7 \times 10^{-3}$ | $5 \times 10^{-3}$ | | | | $5.29 \times 10^{-4}$ |
| | Module 4 | | | | | | | |
| | Module 5 | | | | | | | |
| AMINE | Module 1 | $1.65 \times 10^{-16}$ | $4.9 \times 10^{-11}$ | $2.75 \times 10^{-6}$ | $1.43 \times 10^{-8}$ | $1.41 \times 10^{-5}$ | $2.2 \times 10^{-4}$ | |
| | Module 2 | | | $3.32 \times 10^{-5}$ | | | | $6.20 \times 10^{-10}$ |
| | Module 3 | $4.57 \times 10^{-7}$ | $3.9 \times 10^{-3}$ | | | | $1.29 \times 10^{-5}$ | |
| | Module 4 | $4.57 \times 10^{-5}$ | $3.39 \times 10^{-9}$ | | | | | |
| | Module 5 | | | | | $1.53 \times 10^{-2}$ | $3.42 \times 10^{-14}$ | $1.06 \times 10^{-3}$ |

First two columns list the name of the methods and the identified modules. The following columns contain the false discovery rate associated with hallmarks related to metastatic PDAC cells, that is, epithelial–mesenchymal transition, ECM organization, glycolysis (carbohydrate metabolism) angiogenesis, RAS and PI3K/AKT pathways, and finally, hypoxia.

next used gene set enrichment analysis (Subramanian et al, 2005) to compute enrichments with hallmark and curated terms using annotated genesets from the Molecular Signatures Database (Liberzon et al, 2015). As performed by Chiou et al (2017) on the differential HMGA2+/HMGA2- cell dataset, we searched for terms related to the general metastatic signature of PDAC cancer cells with each module. As expected, we recovered the hypoxic signature already found by Chiou et al (2017). However, contrasting with their analysis, we found evidence for enrichment for previously described gene signatures of metastatic PDAC cells, in many of the five best modules identified by the different methods. The list of all enrichments is presented in Table S2. In Table 1, we specified the false discovery rate (FDR) associated with hallmarks related to metastatic PDAC cells, that is, EMT, ECM organization, glycolysis (carbohydrate metabolism), angiogenesis, RAS and PI3K/AKT pathways, and finally hypoxia (Whittle et al, 2015; Chiou et al, 2017; Wang et al, 2021b). We can see that all methods generate modules that can be associated to one or several metastatic hallmarks (Table 1). Overall, the DOMINO and AMINE methods are the most successful in generating modules associated to metastatic signature. Interestingly, the five modules retrieved by AMINE achieved 17 correspondences matching with metastatic features, whereas only 11 correspondences were identified from the modules generated by DOMINO.

To have a more complete vision of the functions associated with each module, we carried out, for each of them, an enrichment analysis using the facilities offered by the STRING website (https://string-db.org - Szklarczyk et al, 2019). The lists of all enrichments associated with each module with an FDR < 0.05 are presented in Table S3. The first five modules identified by AMINE and their most significant enrichment are presented in Fig 2. The same figures, made to illustrate the results of the methods DOMINO, BioNet, and GiGA are in Figs S7–S9. Results of DIAMOnD are not shown as the method identifies only one module. To ensure that the choice of the annotation associated to each module is not biased, we have systematically shown on the figures the enrichment associated with the lowest FDR among the curated datasets KEGG (Kanehisa et al, 2023), Reactome (Gillespie et al, 2022), WikiPathways (Martens et al, 2021), and Gene Ontology Biological Process (Gene Ontology Consortium, 2019).

It was found that the different methods produced modules with varying degrees of overlap. BioNet was the method that generated the most overlapping modules, whereas GiGA produced perfectly separated modules. Most of the modules generated by all methods clearly corresponded to specific functions or pathways with very low FDR. However, it was noted that for the DOMINO, GiGA, and BioNet methods, some modules showed less clear enrichment, with FDR values above 0.001. In contrast, the AMINE method generated

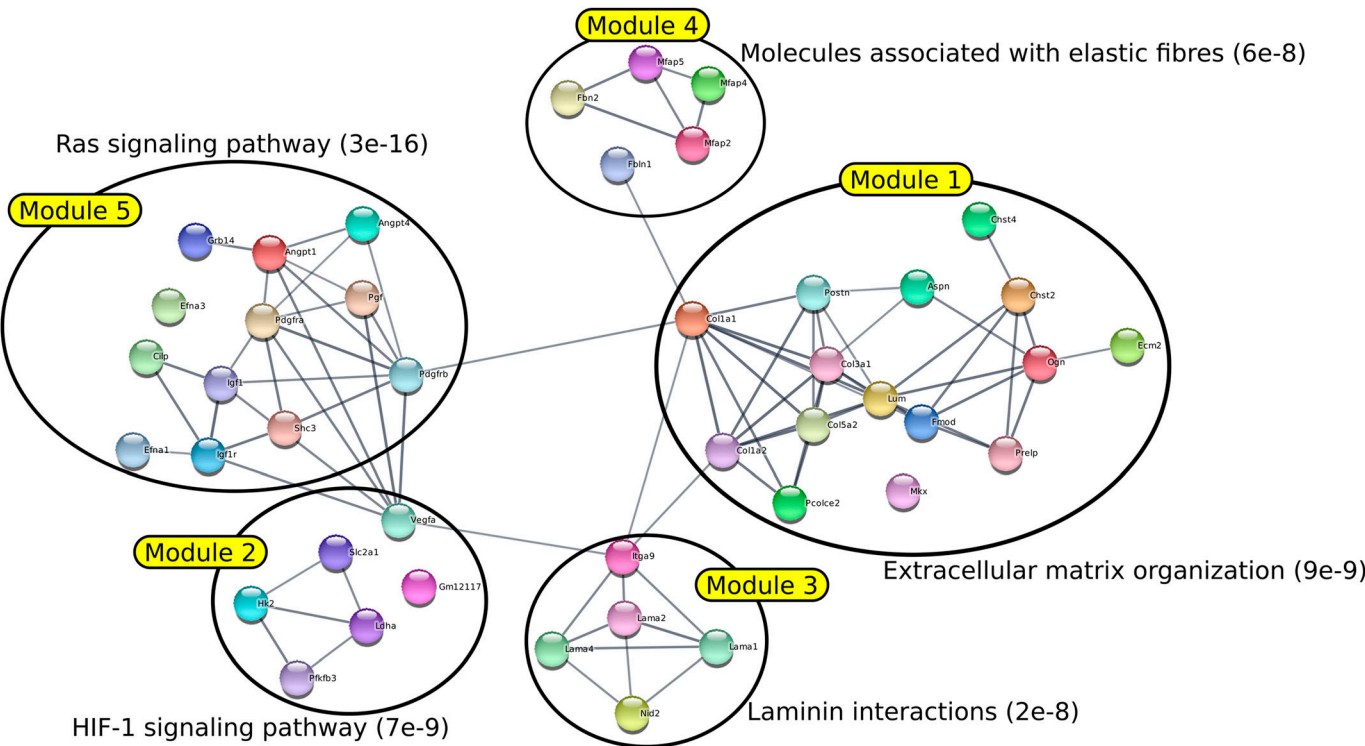

**Figure 2. AMINE reveals modules associated to metastatic process in HMGA2-positive PDAC cells.**
The network was generated by STRING using the five first modules generated by AMINE from the list of deregulated genes in HMGA2+ PDAC cells available in the article of Chiou et al (2017). The annotation of each module is determined by choosing the gene enrichment that is associated with the lowest FDR.

modules that were all enriched with terms associated with an FDR below $1 \times 10^{-7}$. Importantly, the AMINE method was the only one to identify a module specifically associated with the hypoxia pathway. Module 2 was indeed associated with the KEGG "HIF-1 signaling pathway" with an FDR below $1 \times 10^{-8}$. Overall, the comparison of the four methods highlighted the strengths and weaknesses of each approach, with AMINE standing out as particularly effective in identifying enriched modules with very low FDR values and identifying a specific module associated with the hypoxia pathway.

Another strong point of the AMINE method is that it makes it possible to identify a list of modules that is usually very long. As we have shown above, it is then possible to focus on the modules considered as the most significant or, conversely, to specifically target modules associated with some genes of interest. This is what we propose in the following, by analyzing in more detail the results provided by AMINE and, as Chiou et al did in their article, by directing our analysis towards the BLIMP1 gene.

### ECM organization and ECM cell interaction

A hallmark of PDAC is a pronounced collagen-rich fibrotic ECM produced by fibroblasts and cancer cells, known as the desmoplastic reaction. The neoplastic epithelium exists within a dense stroma, which is recognized as a critical mediator of disease progression through direct effects of ECM on cancer cells (Hosein et al, 2020). Notably, three out of the five modules produced by AMINE were found to be linked to the stromal reaction (desmoplasia): modules 1 and 4 were enriched for "organization of the ECM" and module 3 for "cellular interactions with the ECM." Module 1 was more specifically linked to collagen fibril organization, one of the major constituent of PDAC ECM. Indeed, collagen contributes to tumor cell aggressiveness, metastatic process, and chemo-resistance (Shields et al, 2011; Hessmann et al, 2020). Interestingly, module 3 brings together genes involved in the regulation of cancer cell interaction with ECM through focal adhesion kinases and PI3K/ AKT pathways (Fig 2 and Table S3). Based on the available literature, these processes have been strongly involved in the aggressiveness of PDAC cell and the development of metastasis (Jiang et al, 2016; 2020).

### Response to hypoxia

Extensive desmoplasia and hypovascularization within PDAC results in significant intra-tumoral hypoxia (low oxygen) that contributes to its aggressiveness, therapeutic resistance, and high mortality (Koong et al, 2000; Hollinshead et al, 2020). Functional enrichment of modules 2 and 5 raised hypoxia-triggered functions, that is, VEGF- and HIF-1-dependent pathways. These pathways drive angiogenesis, metabolism adaptation of cancer cell to hypoxia (Warburg effect), cell cycle inhibition, enhanced migration, and metastatic progression (Fig 2 and Table S3). These results are in good agreement with the literature on PDAC; for example, these

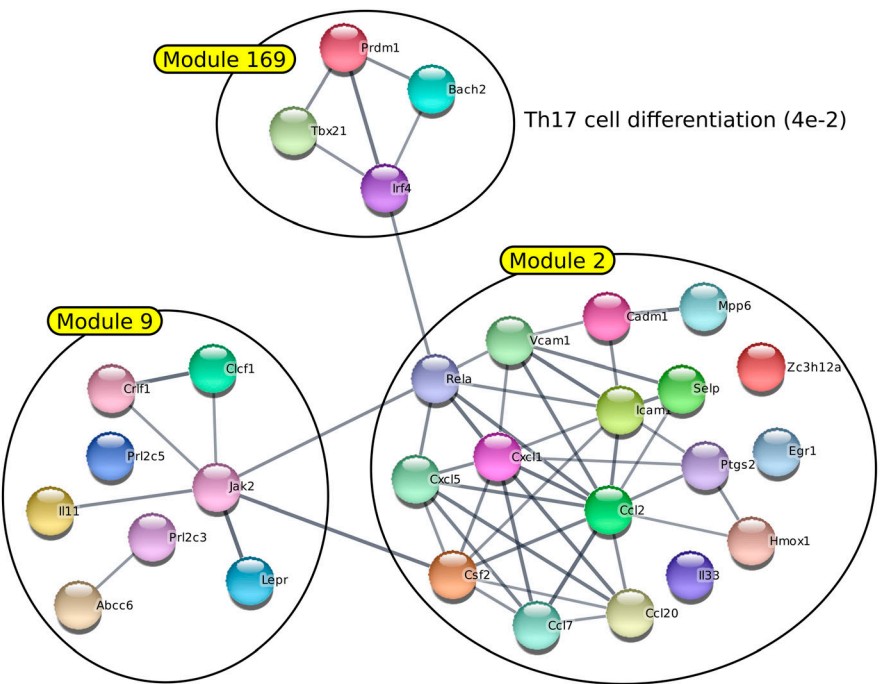

**Figure 3.   BLIMP1 is associated to immune response and inflammation in PDAC cells.**
Module 169 is the BLIMP1-associated module generated by the profiling of genes deregulated in PDAC metastatic HMGA2-positive cell population. Modules 2 and 9 generated by the profiling with AMINE of the genes down-regulated in BLIMP1-silenced PDAC cells. The annotation of each module is determined by choosing the gene enrichment that is associated with the lowest FDR.

pathways are overrepresented in genome-wide transcriptome profiling from ex vivo human PDAC (Ghaderi et al, 2020; Shah et al, 2020). More interestingly, hypoxia-, VEGF-, and HIF-1-associated pathways were repressed in PDAC cells in which BLIMP1, one of the most overexpressed gene in HMGA2+ cell subpopulation, was silenced (Chiou et al, 2017).

Altogether, these results validate our methods because non-oriented analysis of genes deregulated in pro-metastatic PDAC cells by AMINE retrieves gene modules involved in highly relevant functions in the context of the disease.

### In vitro functional validation of BLIMP1-associated module in human PDAC cells

*BLIMP1* is considered a "master regulator" of hematopoietic stem cells, and plays a critical role in the development of plasma B cells, T cells, DCs, macrophages, and osteoclasts. Interestingly, Chiou et al revealed that BLIMP1 is one of the most overexpressed genes in pro-metastatic HMGA2+ PDAC cells (Chiou et al, 2017). The authors analyzed the consequences of *BLIMP1* silencing in mice PDAC cells to unveil its function in disease progression. Based on in silico, in vitro, and in vivo experiments, they concluded that *BLIMP1* acts as a driver of the metastatic ability of PDAC cells. In particular, they found that *BLIMP1* is a hypoxia/Hif-regulated gene in human and murine PDAC which is in a good agreement with functions recovered in modules 2 and 5 raised by AMINE processing (Fig 2 and Table S3). Surprisingly, *BLIMP1* was not included in these modules, but was indeed detected in a module of 4 genes (module number 169 with a *P*-value of 0.044; Table S4 and Fig 3).

Functional enrichment of this module unveiled functions associated to immune response, including regulation inflammation, interleukin production, and Th17 cell differentiation (Fig 3A and Table S5). Interestingly, it is known that neoantigen expression in PDAC results in exacerbation of an inflammatory microenvironment that drives disease progression and metastasis (Hegde et al, 2020). It was therefore tempting to validate this result using a series of functional experiments. In this perspective, we first explored a RNA-Seq experiment performed by Chiou et al revealing deregulated genes in BLIMP1-silenced PDAC cells compared with control (Chiou et al, 2017). AMINE profiling of genes negatively regulated by *BLIMP1* silencing revealed 345 modules with associated *P*-values < 0.05 (Table S6). Among the 10 best modules, we found that modules 2 (*P*-value < $1.01 \times 10^{-11}$) and 9 (*P*-value < $2.83 \times 10^{-8}$) were associated to cytokine production and inflammatory process (Fig 3 and Table S5) after functional enrichment. Next, to confirm the putative involvement of BLIMP1 in epithelial cancer cell inflammatory process in vitro, we silenced *BLIMP1* in MIA PaCa-2 cells, a human PDAC cell line, using siRNA silencing, (Fig 4A), and explored how it modified the profile of cytokine secretion using a cytokine profiling array and Western blot experiments. Indeed, we found that BLIMP1 repression triggered the production of IL-18Bpa and angiogenin, two anti-inflammatory factors (Lee et al, 2014) and reduced the secretion IL-6, a major pro-inflammatory interleukin (Tanaka et al, 2014) (Fig 4B and C).

Altogether, these results indicate that the participation of blimp-1 in inflammatory process predicted by AMINE could be confirmed in vitro. They further validate AMINE as a valuable method to detect relevant functional modules from large experimental datasets. Our study therefore unveils a new function of BLIMP1, in the regulation

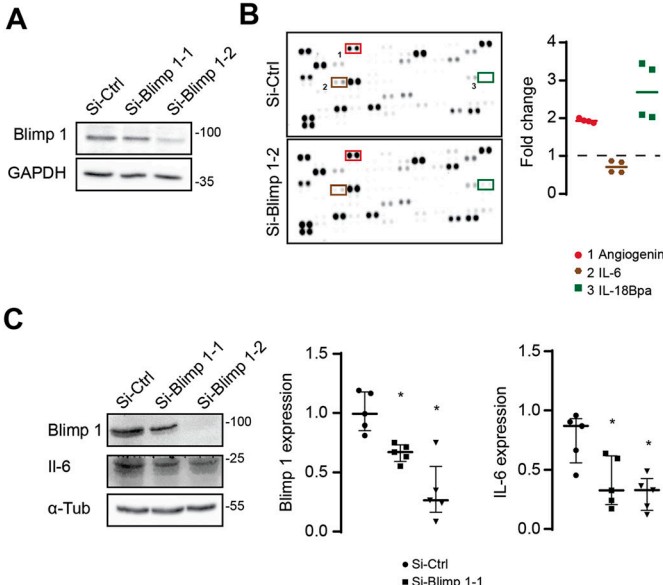

**Figure 4. BLIMP1-silencing modifies the cytokine secretion profile in PDAC cells.**
**(A)** Immunoblots of Blimp 1 in MIA PaCa-2 cells transfected with a non-targeting siRNA (Si-Ctrl) or with two different siRNA targeting Blimp-1 (Si-Blimp 1–1 and Si-Blimp 1–2). Data are representative of three independent experiments. **(B)** Soluble cytokine protein expression was assessed using cytokine arrays in si-Ctrl or si-Blimp 1 transfected MIA PaCa-2 (N = 2; n = 4). Representative arrays are shown. On the left panel, values from densitometry quantification are shown as a fold change from the control. **(C)** Immunoblots of Blimp 1 and IL-6 in MIA PaCa-2 cells transfected with a non-targeting siRNA (Si-Ctrl), or with two different siRNA targeting Blimp-1 (Si-Blimp 1–1 and Si-Blimp 1–2). On the left panel, values from immunoblots densitometry quantification are shown as scatter plots, n = 5, * = P < 0.05.

of PDAC-related inflammatory process triggered by tumoral epithelial cells.

# Discussion

This article proposes a new method for identifying gene modules that are activated as a result of a state shift caused by a biological experiment. Our method, called AMINE, uses as inputs, on the one hand, the ultimate result of any RNA-Seq analysis pipeline which is the differential expression of genes, and on the other hand, a network modeling the interactions between genes.

Although many methods have been developed over the past two decades, AMINE stands out for its ability to accurately identify modules on datasets designed to mimic the structure of biological networks, outperforming many other competing methods. Extrapolating these results to a measure of accuracy on real datasets is very difficult. There is no method to ensure that good predictions on artificial data translate into good predictions on real datasets. However, we have made a special effort to ensure that our simulations are close to real datasets. The networks we generate, with the parameters presented in this article, are closer to a real interaction network than the networks used by some competing methods. In addition to the results reported in this article, several

studies utilizing AMINE to analyze various types of data have already been published (Feliz Morel et al, 2022; Pasquier & Robichon, 2022a; 2022b), which also emphasizes the relevance of the results obtained by the method.

Several studies utilizing AMINE to analyze various types of data have already been published (Feliz Morel et al, 2022; Pasquier & Robichon 2022a; 2022b).

It is known that the interaction networks stored in public databases are both incomplete and contain erroneous interactions. In their study, Von Mering et al (2002) estimate that, for *Saccharomyces cerevisiae*, the protein–protein interaction data (PPI) reported in public databases account for only one-third of existing interactions. This observation suggests that methods relying heavily on network topology may not be very suitable. In particular, the effectiveness of methods based on clique identification may be questionable if we consider that a significant proportion of protein–protein interactions remain unknown.

This observation leads us to believe that methods based very precisely on the topology of networks are not to their advantage. We think first of all of the methods based on the identification of cliques. Their performance is more than questionable if we consider that a large part of the interactions between proteins are unknown. If we consider that the missing interactions are randomly distributed on the graph, we can estimate that all the paths on the graph are impacted in the same way and thus that the methods based on random walks could be the least affected. Intuitively, we can indeed argue that, in a graph on which a certain proportion of the edges have been randomly deleted, if, from a source node A, random walks allow on average to reach node B before node C, then, on the complete graph, node B will probably always be closer to node A than node C. The other problem with methods based on graph traversals is that PPI networks are "small-world" networks, meaning that the neighbors of a given node are likely to be neighbors of each other, and most nodes can be reached from every other node by a small number of hops. Thus, any method that relies on graph traversals will find that a large portion of the network is close to any typical node (Cao et al, 2013). We indeed argue that network-embedding methods can provide the backbone of reliable methods by estimating distances between nodes that take into account the entire topology of the graph and, moreover, are little affected by the proportion of missing edges. Our original method works on an embedding of an interaction network by adopting a greedy algorithm and an active subnetwork relevance measure defined in other articles (Ideker et al, 2002). The great advantage of our method is that it does not require any parameterization; it is not even necessary to indicate the number of modules to be identified or the size of the modules.

We have checked that our method performs well on artificial datasets and compares favorably with existing methods that are the current state of the art. We then processed a real dataset from a study focused on PDAC, on which AMINE retrieved modules associated with functions recapitulating PDAC metastatic process such as the response to hypoxia and extra-cellular matrix-dependent signalling. Moreover, our studies show that AMINE can identify modules corresponding to functions not revealed by traditional approaches consisting in analyzing only the most differentially expressed genes. Indeed, we found that *BLIMP1,* one of the most

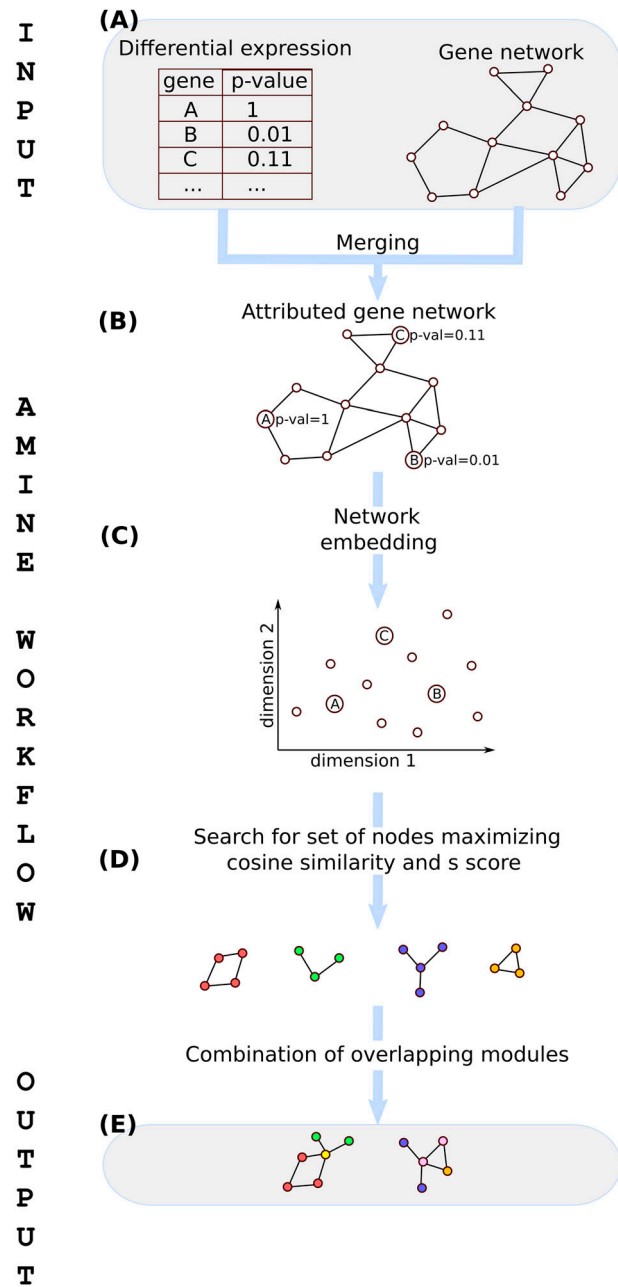

**Figure 5. Workflow of the AMINE method.**
**(A)** Input data are composed of a table storing the significance of the expression variation of genes between two conditions and a network representing known gene interactions. **(B)** Data about gene interactions and gene variations are merged to generate an attributed gene network. **(C)** Nodes belonging to the attributed gene network are mapped to a low-dimensional space through the use of a biased Node2Vec method. **(D)** Sets of genes that are both cohesive and differentially expressed are identified in the embedded space by maximizing both the scores of the nodes and the cosine distance between the vectors representing the nodes. **(E)** Redundancy in the content of modules is ruled out by combining sets of nodes obtained in the previous step while ensuring that the result remains spatially cohesive.

up-regulated genes in highly metastatic cells, was included by AMINE in a module ontologically associated to pro-inflammatory process, a result confirmed in vitro. Indeed, the silencing of *BLIMP1* in human

PDAC cells altered cytokine production. However, we stress that our method is not an alternative to methods based on the identification of the most differentially expressed genes, but rather a complement to these approaches.

In this article, we used the STRING protein–protein interaction network for mouse. However, we would like to emphasize that AMINE is not limited to these data. The method is flexible and allows users to choose between the STRING, BioGRID, and IntAct PPI networks for four different organisms, namely *Caenorhabditis elegans*, *Drosophila melanogaster*, *Homo sapiens*, and *Mus musculus* and it is possible to easily add others. In addition, users have the possibility to upload their own network in a simple format, which would allow using AMINE with other types of biological networks, such as metabolic or gene-regulatory networks. Using these additional networks in AMINE, potentially by combining them, could provide further insights into biological systems.

## Materials and Methods

The AMINE method predicts active modules from data consisting of background knowledge about gene interactions and measurements representing, in the specific context of a given experiment, indicators of the involvement of genes in the studied process. This concept of gene involvement is materialized by a *P*-value which quantifies, for each gene, the statistical significance of its variation (Fig 5A).

Data about gene interactions and gene variations are merged to generate an attributed gene network in which genes are annotated with a numeric attribute representing the extent of their variation (Fig 5B). Mathematically, the dataset is represented as an attributed graph $G = (V,E,\lambda)$ consisting of a set of vertices (also called node, that symbolizes the genes), a set of edges $E \subseteq \{(u,v) \in V^2 \vee u \neq v\}$ and a value function $\lambda(v):V \rightarrow R$ which associates a value $P \in R$ to each vertex $v \in V$. An induced subgraph of $G$ is a subset of the vertices of $G$ together with those edges of with both endpoints in $S$. Many active module detection algorithms focus on identifying induced subgraphs whose values associated with their nodes stand out from the values associated with the other nodes of the graph. We hypothesize that focusing heavily on the detection of connected sets of genes may not be optimal, given the fact that the interactions between genes described in the databases are still largely incomplete. For this, we adopt a definition of a module that is closer to the one used in cluster analysis: objects that are grouped together (in a module) are more similar to each other than to those in other groups. The notion of similarity encompasses a component taking into account the distance on the graph between the vertices belonging to a module and a significant nearness between the values associated to these vertices.

### Scoring of a subgraph

Let $P_i = \lambda(v_i)$ be the associated *P*-value of vertex $v_i$. We aggregate the *P*-values associated to the nodes of a subgraph with Stouffer's Z method, the same strategy used by Ideker et al (2002). If we let $z(v_i) = \Phi^{-1}(1-\lambda(v_i))$, where $\Phi$ is the standard normal cumulative distribution

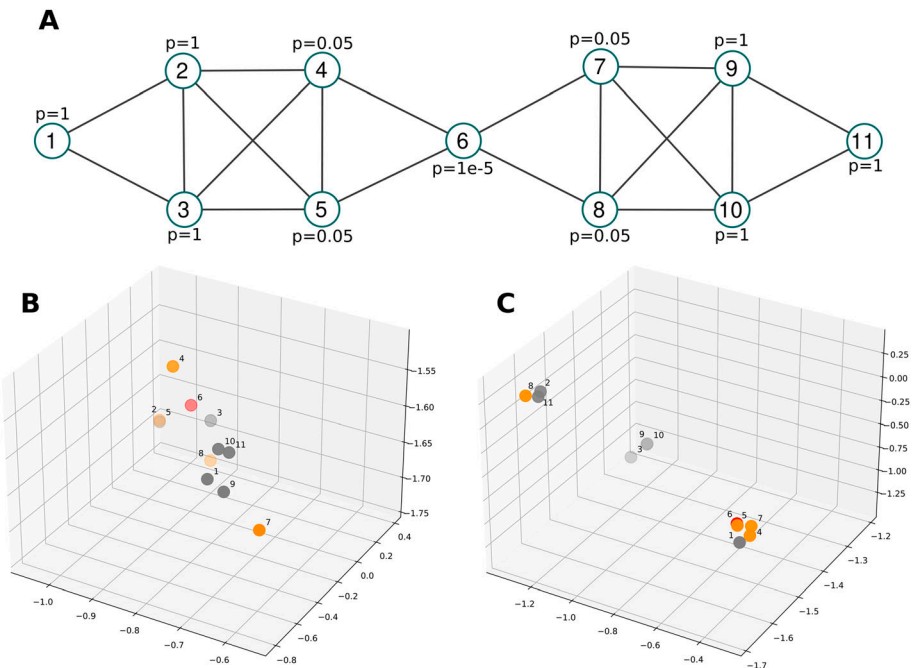

**Figure 6. Illustration of the impact of using a biased random walk on the embedding based on a toy example.**
**(A)** Illustration of a toy graph composed of 11 nodes. Each node is associated with a *P*-value (named p in the figure) that goes from 1 × 10⁻⁵ for the central node to 1 for the peripheral nodes through 0.05 (the minimal *P*-value usually used to consider that a variation observed is effective). **(B)** Standard embedding obtained by node2vec. Nodes are spread out and their position roughly reflects their location on the graph. Indeed, with the exception of node 1, nodes 2–6 are on one side and nodes 7–11 on the other. **(C)** Embedding obtained using our proposed biased random walk that embeds the similarity of node values. Here, the figure shows distinct clusters and the foreground one, which contains nodes 1, 4, 5, 6, and 7, represents a satisfactory grouping that contains a number of nodes if interest (appearing in orange and red).

function, then, the aggregate z-score $z_a(G')$ for an induced subgraph $G' \subseteq G$ composed of $k$ vertices, is computed with

$$z_a(G') = \frac{1}{\sqrt{k}} \sum_{v_i \in G'} z(v_i).$$

To get a subnetwork which has higher aggregation z-score compared with a random set of vertices, we define, still following the same methodology as Ideker et al (2002), a corrected score $s(G')$ of a subgraph with

$$s(G') = \frac{z_a(G') - \mu_k}{\sigma_k},$$

where the mean $\mu_k$ and SD $\sigma_k$ are computed based on a Monte Carlo approach, taking 10,000 rounds of randomly sampling a connected subgraph of $k$ vertices from $V$. From $s(G')$, we can easily compute the probability of observing in $G$, a subnetwork of the same size as $G'$ with a corrected score at least as extreme as the one observed. This is given by the one-sided *P*-value: *P-value*$(G') = 1 - \Phi(s(G'))$.

### Network embedding

A network-embedding method is a function $\psi: V \to R^m$ that associates to each vertex $v$ of the graph a vector $d$ of size $m$. Node2vec (Grover & Leskovec, 2016) uses a biased random walk procedure which efficiently generates diverse neighborhoods of a given node. Node contexts are then processed with the word2vec method (Mikolov et al, 2013). Node2Vec uses two parameters to control the walks. Intuitively, these parameters control how the walk explores and leaves the neighborhood of starting nodes. They allow a tuning between outward exploration and local walking.

However, in our case study, it may indeed be interesting, instead of using biases that only considered the topology of the network, to use the data associated to nodes, that is, the value of *P*. The idea is to bias the walk so that when the walker is located on a node, transitions to nodes with similar values of *P* (Fig 5C) are favored. As *P* represents a *P*-value, the walker will be encouraged to favor visits of correlated and anti-correlated genes. We have conducted many experiments by replacing the parameters proposed by Mikolov et al (2013) with our suggested use of similarity between nodes or by combining the different ways to bias the walk. It turns out, in the end, that using only the bias based on the similarity of *P*-values gives the best results. The bias we introduced allows to control the walk by assigning a transition $t$ from a node $i$ to a node $j$ proportional to $t_{i,j} = max(1 - |P_i - P_j|, e)$ with $e$ being a very small value (concretely set to 1 × 10⁻¹⁶ in the algorithm) that prevents obtaining a transition probability between two nodes equal to zero.

Other parameters tuning the Node2Vec method are given below:

- Number of walks: 20
- Walk length: 100
- Vector dimensions: 128
- Window size: 5
- Epoch: 10.

### Assessing the impact of using biased random walk on embedding

To test the ability of the biased random walk to take into account the weight of the nodes and to visualize the changes that this brings on the embedding, we conducted an experiment on a toy example. Fig 6A shows a visualization of a toy graph composed of 11 nodes. Each node is associated with a *P*-value that goes from 1 × 10⁻⁵ for the central node to 1 for the peripheral nodes through 0.05 (the

minimal *P*-value usually used to consider that a variation observed is effective). This graph is topologically composed of two modules (nodes 1–5 and nodes 7–11) with a central node which is node 6. We are interested in the changes that the bias we introduced produces on the embedding. To do so, we generated a standard embedding obtained by node2vec (Fig 6B) and an embedding obtained using our proposed biased random walk (Fig 6C) with only three dimensions to be able to visualize it.

We observe that the nodes of interest, appearing in orange and red, are further clustered using a biased random that embeds the similarity of node values. Furthermore, in the standard embedding (Fig 6B), the nodes are spread out, whereas Fig 6C shows distinct clusters. The foreground cluster contains nodes 1, 4, 5, 6, and 7 which form a satisfactory grouping, although node 1 is out of place and node 8 is in the upper-left corner cluster containing nodes 2, 8, and 11. These results demonstrate that the embedding works as expected and will allow for the identification of modules of interest much more easily than with an embedding solely based on network topology as produced by standard node2vec method.

### Algorithm

The cohesion measurement of a set of nodes on the graph can be determined on the embedded space using the cosine distance *cos* between the vectors representing the nodes. We use this property to identify the most similar nodes to a given node $v_i$ (represented as

---

**Algorithm 1:** Active modules identification

**Input:** a node valued graph $G$
**Output:** a set of modules, represented as sets of nodes

1 $M \leftarrow \varnothing$ ▷ *set of active modules*
2 $E \leftarrow \text{Node2Vec}(G)$ ▷ *network embedding of G*
3 **for** $v_i \in G$ **do**
4 $M_i \leftarrow \{v_i\}$ ▷ $i^{th}$ *element of M*
5 $score \leftarrow s(M_i)$
6 $C_i \leftarrow sort_{cos(v_i,v_k)}(\{\forall v_{k \neq i} \in E\})$ ▷ *descending sort using*
 ▷ *cosine distance to* $v_i$
7 $j \leftarrow 1$
8 **while** $j \leq |C_i|$ **and** $s(M_i \cup \{C_{i,j}\}) > score$ **do**
9 $M_i \leftarrow M_i \cup \{C_{i,j}\}$ ▷ *add* $j^{th}$ *vertex of* $C_i$ *to* $M_i$
10 $score \leftarrow s(M_i \cup \{C_{i,j}\})$
11 $j \leftarrow j + 1$
12 $sort_s(M)$ ▷ *descending sort of modules using s score*
13 $BM \leftarrow \varnothing$ ▷ *best active modules*
14 **for** $i = 1$ **to** $|M|$ **do**
15 $s_{best} \leftarrow s(M_i)$ ; $BM_i \leftarrow M_i$
16 **for** $k \in ]i, |M|]$ **do** ▷ *for all other modules*
17 $v_k \leftarrow M_{k,1}$ ▷ *1st element of* $k^{th}$ *module*
18 **if** $v_k \in M_i \wedge s(M_i \cup M_k) > s_{best}$ **then** ▷ *merging modules ?*
19 $s_{best} \leftarrow s(M_i \cup M_k)$
20 $BM_i \leftarrow M_i \cup M_k$
21 $M \leftarrow M - \{M_k\}$
22 **return** $BM$

---

**Figure 7. Algorithm of AMINE.**
The algorithm consists of two main stages. In the first phase (lines 3–11), a greedy approach is used to assign to each node in the graph, the set of neighboring nodes in the embedded space that maximizes the s score. The aim is to iteratively extend each cluster as long as the score continues to increase. The second phase (lines 14–21) involves combining the individual clusters while ensuring that the merged clusters maintain spatial cohesion.

similar function in the algorithm summarized in Fig 7). Thanks to a greedy approach, we collect, from each node, clusters $M_i$ of increasing size evaluated using the s score previously defined. Our strategy is to expand the cluster as long as the s score increases (lines 8–10 of the algorithm). In practice, as we are very strict on the stopping condition, the clusters obtained are quite small (usually 5 nodes at most). At the end of this phase, we obtain a list of clusters, each one centered on a node, with each cluster being assigned a corresponding s score (Fig 5D). The cluster centered on vertex $v_i$ is thus denoted $M_i$ with $M_i \subseteq V$ and $v_i \in M_i$.

The next step of the method consists in combining the different clusters while ensuring that the new merged clusters retain spatial cohesion (Fig 5E). In this context, we say that two clusters $M_i$ and $M_j$ are spatially cohesive when there is a meaningful intersection between them. Concretely, we do not simply rely on the presence of overlapping nodes, but assert that a cluster $M_j$ is cohesive with a cluster $M_i$ only if its center $v_j \in M_i$. Cluster aggregation consists in processing one by one the clusters found in the previous step, starting from the cluster with the highest s score (line 13 of the algorithm), evaluating the clusters formed by the union of $M_i$ with one or more cohesive clusters and keeping the resulting clusters with the highest s scores (lines 15–19). The workflow and the algorithm of the AMINE method are presented in Figs 5 and 7.

### Parameters used with other methods

For all methods, we used the standard parameters used in the reference articles. DIAMOnD and DOMINO require a list of seed genes (i.e., differentially expressed) as input. The choice of the number of these seed genes is not specified in the manuscripts presenting the methods and is left to the discretion of the user. From a list of n genes associated with adjusted *P*-values, we used as seed genes all genes with an adjusted *P*-value < 0.001/n, as done by Lazareva et al (2021) in their article. GiGA needs as parameter the maximum size of the identified modules. We executed the method specifying a size of 20, which is the default. DIAMOnD needs as parameter the exact size of the module to be found. The default value is 200 but we consider this to be far too high. To be able to compare the results with other methods that return small modules, we chose a size of 20, as for GiGA.

### Generation of realistic interaction network

We use an extended version of the Barabasi-Albert model of preferential attachment (Albert & Barabási, 2000), to generate several artificial networks by varying the parameters p and q controlling the probabilities to add and remove edges respectively and the parameter m specifying the number of initial nodes. Our results suggest that using three initial nodes with parameters p and q set to 0.09 and 0.70, respectively, allows to generate random network with topologies relatively close to real interaction networks (Table S7).

1,000 graphs were generated using these parameters (an example of this kind of graph is given in Fig S3). The strategy to specify the value of nodes is exactly the same at the one used by Robinson et al (2017). To be able to do a comparison with other methods, we generated only one module of designated hits. As AMINE is

dedicated to the identification of relatively small modules (to focus on really relevant genes that can be investigated by biologists), we have targeted our tests on the identification of small modules of sizes 10 and 20.

### Selection of the network for analyzing real datasets

All experiments were performed using the mouse PPI STRING network with interactions having a global confidence score higher than 0.7, as is usual.

In the GitHub repository (https://github.com/claudepasquier/amine), we provide access to three public sources of PPI networks, namely STRING (Szklarczyk et al, 2019), BioGRID (Oughtred et al, 2021), and IntAct (Del Toro et al, 2022), for four organisms, namely *C. elegans*, *D. melanogaster*, *H. sapiens*, and *M. musculus*. We also provide a file that represents the union of these three networks. In addition, users can upload their own PPI network and associated *P*-values using two simple files. We provide examples of these files so that users can easily adapt their own network for use with AMINE. The code also includes options for filtering PPI data. For example, for STRING, it is possible to filter the data based on each component of the confidence score. By default, we use a filter that only includes interactions with an overall confidence score greater than 0.7. For BioGRID, users can filter interactions based on the type of interaction (physical or genetic). By default, all interactions are used. For IntAct, it is possible to filter based on a minimum value of the confidence value. By default, all interactions are retrieved.

In addition to the freely distributed sources that allow a user to run AMINE locally, we provide access to a website where users can run the application without installing anything on their machine. With this version (available at http://amine.i3s.unice.fr), users can analyze their datasets using the PPI networks for the four previously mentioned organisms. As with the experiments, only interactions with a confidence score greater than 0.7 are used. Thus, a user only needs to provide the differential gene expression data generated by the pipeline of his choice. From a very simple interface (Fig S10), he only has to specify the name of the organism analyzed, the file on which the data are located, and the ID of the columns containing the genes' names, the *P*-values, and optionally, the fold changes to be able to launch the process. The address of the page containing the results is e-mailed to the user when the processing is completed. On the result page, the most significant modules are listed, however, all the modules found can be downloaded as an Excel document consisting of two sheets. The first sheet, named "list of modules" contains the list of all modules found. The results are presented in 4 columns containing the module number, the list of genes in the module, the s score of the module, and the associated *P*-value. The second sheet, named "genes to modules," is composed of two columns: the first one contains the name of a gene and the second one, the module to which it belongs.

It should be noted that we designed this website to allow users to easily use our application without installing anything locally. However, the resources are limited, both in terms of CPU and bandwidth. It is not intended for intensive use. We strongly advise users who wish to maximize the potential of AMINE to install the application locally on their machine.

### Evaluating the algorithm's resistance to noisy data

To assess the impact of the network used by the AMINE method, and in particular, to provide some insights on the robustness of the method to noisy data, we analyzed the Chiou et al data using different interaction networks, including BioGRID, Intact, STRING with a global score threshold set to 0.9 and 0.4 (in addition to the threshold of 0. 7 in the article) and STRING applying a threshold of 0.7 on the different components of the global score, that is, co-expression, database, experimental, and text mining (the filtering on neighborhood, fusion, and co-occurrence sources generate networks too small and sparse to be successfully used to identify a module). The modules identified using each of these configurations are presented in Tables S8–S15, and the best annotation found for each module is presented in Table S16. It can be seen that depending on the type of network used, the enrichments differ. However, the same subset of relevant annotations, such as those related to glycolysis and gluconeogenesis, ECM organization, and elastic fiber formation, are generally found.

Looking at the number of enrichments obtained with an FDR < $1 \times 10^{-5}$ for each network, we see that the different versions of the STRING database filtered on the basis of the global score are those with which the top 5 modules obtain an annotation. STRING filtered on the database score also sees 5 of these modules enriched. STRING filtered with co-expression and text mining scores sees 4 of the first five modules enriched. STRING filtered by considering only experimental data associated with a score > 0.7 is the network that gets the least significant enrichment, as only 2 out of 5 modules have enrichments associated with an FDR < $1 \times 10^{-5}$. Regarding the other databases, the results obtained seem less relevant than with STRING because with BioGRID and Intact, only two modules are annotated with an FDR < $1 \times 10^{-5}$. One of the most relevant annotations in the context of the Chiou et al study is the pathway directly associated with hypoxia: "HIF-1 signaling pathway." This annotation is only identified with STRING filtered on the basis of a global score higher than 0.7 and higher than 0.4.

We can draw two main observations from this experiment. Firstly, the AMINE method performs well regardless of the network used and appears to be tolerant to noisy data, such as that derived solely from text mining, as relevant annotations are obtained from this dataset, consistent with those obtained using more complete data. Secondly, using the STRING database by default with a threshold of 0.7 on the global score appears to be relevant, as it allows for similar enrichment results to those obtained using a threshold of 0.4 with a smaller dataset, thus reducing the algorithm's execution time.

### Cell culture

The PDAC cell line, MIA PaCa-2, was obtained from Richard Tomasini CRCM, Marseille, France, and culture in DMEM (Gibco, Life Technologies Limited) supplemented with 10% FBS, and penicillin/streptomycin. Cells were maintained at 37°C in a humidified atmosphere (5% $CO_2$). The cells were tested routinely for mycoplasma contamination.

### siRNA transfection

siRNAs (Sigma-Aldrich) were used for BLIMP 1 silencing. Non-targeting (si-Ctrl: SIC001) or BLIMP 1-targeting siRNAs (si-Blimp 1-1: 5′CUUGGAAGAUCUGACCCGA-3′; si-Blimp 1-2: 5′CCUUUCAAAU-GUCAGACUU-3′ were transfected in MIA PaCa-2 cells using Lipofectamine RNAiMAX (Invitrogen, Life technologies Corp.) following the manufacturer's instructions. The final siRNA concentration was 30 nM. The medium was changed 8 h after transfection and the efficiency of the transfection was assessed by Western blot after 72 h.

### Western blotting

Cells were lysed in RIPA buffer supplemented with Complete Protease Inhibitor Cocktail and PhosSTOP Phosphatase Inhibitor Cocktail (Roche Diagnostics GmbH). Lysate were centrifuged at 15,294$g$ for 15 min at 4°C and then protein concentration was quantified using Bradford assay. Protein lysates were subject to SDS–PAGE and transferred onto a PVDF membrane. The membranes were blocked with 5% low fat milk in Tris-buffer saline–tween (TBS-T) for 1 h. The membranes were incubated in Blimp 1 antibody (diluted at 1:1,000; Cell Signaling) overnight. The membranes were washed in TBS-T followed by incubation with horseradish peroxidase-conjugated secondary antibody for 1 h at room temperature (Sigma-Aldrich). The signal was then visualized using ECL reagent (Immobilon Western, Millipore) and chemoluminescence detection system (fusion FX7 Edge; Vilber).

### Human cytokine array

For the cytokine assay, the Proteome Profiler Human XL Cytokine Array Kit (R&D Systems) was used. The array was carried out using 500 $\mu$l of cell supernatants obtained by incubating MIA PaCa-2 in DMEM 0% FBS, 48 h after siRNA (si-Ctrl or si-Blimp 1-2) transfection, following the manufacturer's instructions. For analysis of cytokine arrays, the intensity of each spot was measured using ImageJ software. The background was removed from all values, and they were normalized to the positive control spots.

### Statistical analysis

Results are presented as median with interquartile range unless stated otherwise. Kruskal–Wallis tests followed by a Dunn's post test were used to compare data. Analyses were performed using GraphPad Prism V.8.0.1. A $P$ value < 0.05 was considered statistically significant.

## Data Availability

All data generated or analysed during this study are included in this published article and its supplementary information files. In addition, synthetic data generated by Robinson et al (2017) are available for download on the GitHub repository of AMINE, specifically in the "data/synthetic" directory. The artificial data generated for this project are also regenerable using the generator included in the code deposited on GitHub. RNA-Seq data GSE90625 and GSE90824 from Chiou et al (2017) are accessible at https://www.omicsdi.org/dataset/geo/GSE90625 and https://www.omicsdi.org/dataset/geo/GSE90824, respectively, and have been copied onto the GitHub repository under the directory "data/real/expression/chiou_2017." Differential expression analyses of these data were performed with the DESeq2 R package with default configuration, and the resulting output has been stored in the same directory.

## Supplementary Information

## Acknowledgements

The authors would like to acknowledge Camille Bérenguier, who critically proofread the article. They are grateful to the OPAL infrastructure from Université Côte d'Azur and the Université Côte d'Azur's Center for High-Performance Computing for providing resources and support. This work was supported by the French government, through the UCA[JEDI] Investments in the Future project managed by the National Research Agency (ANR) with the reference number ANR-15-IDEX-01, the French National Research Agency through the LABEX SIGNALIFE program (ANR-11-LABX-0028-01), La Ligue contre le Cancer (GB/MA/IQ-10607), and Fondation ARC (PJA 2016 120 4740, PJA 2019 120 9546 and PJA 2018 207701).

### Author Contributions

C Pasquier: conceptualization, data curation, software, supervision, validation, investigation, visualization, methodology, and writing—original draft, review, and editing.
V Guerlais: data curation, software, and visualization.
D Pallez: validation, investigation, visualization, methodology, and writing—original draft, review, and editing.
R Rapetti-Mauss: resources, investigation, visualization, methodology, and writing—original draft, review, and editing.
O Soriani: conceptualization, resources, supervision, validation, investigation, visualization, methodology, and writing—original draft, review, and editing.

### Conflict of Interest Statement

The authors declare that they have no conflict of interest.

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
