## [Reviewer comments · Life Science Alliance]

Life Science Alliance

A network embedding approach to identify active modules in biological interaction networks

Claude Pasquier, Vincent Guerlais, Denis Pallez, Raphaël Rapetti-Mauss, and Olivier Soriani

DOI: <https://doi.org/10.26508/lsa.202201550>

Corresponding author(s): *Claude Pasquier, French National Centre for Scientific Research*

Review Timeline:

Submission Date:	2022-06-07
Editorial Decision:	2022-07-11
Revision Received:	2023-04-25
Editorial Decision:	2023-05-25
Revision Received:	2023-06-06
Accepted:	2023-06-06

Scientific Editor: Novella Guidi

Transaction Report:

July 11, 2022

Re: Life Science Alliance manuscript #LSA-2022-01550-T

Dr. Claude Pasquier
French National Centre for Scientific Research
Laboratoire I3S
CS 40121
Sophia Antipolis 06903
France

Dear Dr. Pasquier,

Thank you for submitting your manuscript entitled "A network embedding approach to identify active modules in biological interaction networks" to Life Science Alliance. The manuscript was assessed by expert reviewers, whose comments are appended to this letter. We invite you to submit a revised manuscript addressing the Reviewer comments.

Thank you for this interesting contribution to Life Science Alliance. We are looking forward to receiving your revised manuscript.

Sincerely,

B. MANUSCRIPT ORGANIZATION AND FORMATTING:

Reviewer #1 (Comments to the Authors (Required)):

In this study the authors suggest a network embedding based active module identification method. In this context modules are not constrained to be connected components but they are selected to be clusters in the embedding space. Network embedding is performed using biased random walk, in which authors defined their own bias: The transition probability between two nodes is high if their p-values are close to each other.

*** Main concerns ***

* The codes of AMINE should be shared on GitHub.

* In Data Access section it is stated that "All data generated or analysed during this study are included in this published article and its supplementary information files."

I do not have access to these files, will they be available in the final publication?

* AMINE was compared with a baseline method and three active module identification methods. The reason of choosing these tools for comparison is not justified. There are more recent methods, such as DOMINO (<https://doi.org/10.15252/msb.20209593>). DOMINO was shown to be performing very well in its publication also in a review study (<https://doi.org/10.1093/bib/bbab066>). It is important to see the performance comparison of these methods, whether it is in favor of AMINE or not.

* AMINE was compared with other methods only using artificial datasets. On real data, AMINE was evaluated by the true positive enrichment analysis results of the modules. The same can be done with the compared methods.

* In the enrichment analysis results of the modules obtained using real data (Table 1) it is not clear whether the annotation terms are the most significant ones for those modules. How many other terms were enriched for each module? Could you comment on the precision of the enrichment results? I know that there is no ground truth, so I am not expecting a numerical value but similar to the way true positive enrichment results are commented on, a rough interpretation of precision can be done.

* AMINE does not constrain connectivity in the modules. To present how connected the nodes in each module are, can you plot edge density (number of edges / number of possible edges) for each module on a boxplot (as a supplementary file)?

* In active module identification, the selection of PPI network is obviously very important. Researchers usually favor a network over others based on how the interactions were identified or curated. In AMINE the user does not have an option to use another PPI. The user should have the option to upload their PPI or use the provided one.

* A common approach while using STRING database is to filter interactions by confidence level (e.g. selecting the interactions with confidence score greater than 0.7) or source (e.g. excluding text-mined interactions). Did you apply any filtering?

* The bias in RWR that is introduced by the authors (defined in Methods - Network embedding section) should be discussed. In active module identification, if a node is connected to significant nodes, then there is a chance that it is involved in the module ("guilt by association"). However in the approach followed here, nodes with similar level of significance will be more similar in the vector representation, thus they will be clustered, which will decrease the chance of an insignificant node being involved in the module.

* On <http://amine.i3s.unice.fr/>, there is no initial checking of the submitted files (I tried some files). If there is a problem in the input file the user should be notified early.

* On <http://amine.i3s.unice.fr/>, with each consecutive submission (submitted before the previous one ends) waiting time increases one hour, seems like one job can be run at a time. In this state, making people use this tool might be hard.

*** Minor concerns ***

**Introduction

* In this study the literature on active module identification is very well presented. However there is not a mention of any other study that uses network embedding to prioritize genes/proteins (e.g. <https://doi.org/10.3389/fgene.2019.00226>). The literature on this subject should also be mentioned.

* "Active module" generally refers to a set of connected nodes, here there is no constraint for connectedness. This is not mentioned until the Methods section, please make your definition of module earlier (in Introduction).

* Lines 38-40: The sentence "In order to ..." is hard to read/understand at first sight, it might be improved.

* Lines 53-54: "These pertinent subnetwork are" -> "subnetworks"

* Lines 54-55: Some of the names are plural, some are singular, it will be better if they are consistent.

* Line 57: "one based on a measurement of genes activity" -> "one based on the measurement of genes' activities"

* Line 69: It is not clear what it is meant with distance (distance on network or opposite of similarity), also it is not clear why relying on distance is bad.

* Lines 73-88: There is a problem with the story flow here, at line 80 there is Node2Vec, but then deep learning and word embedding came... Shouldn't word embedding be mentioned before Node2Vec, if necessary?

* Lines 97-99: "a greedy approach to build increasingly large clusters of nodes". As a reader who knows how greedy approach works, this sentence made me think like there is an extra process to the greedy approach, like several clusters with different sizes were built, however it is just the greedy approach expanding the cluster. I suggest you to reconsider this sentence. Another point, the sentence is like the greedy approach is used to evaluate the clusters. Could it be like "Our method uses a greedy approach to build the clusters based on the similarity of the nodes' encoding vectors and a metric that takes into account the activity of the contained nodes." ?

* Line 102: "AMINE outperforms 5 recent methods". 5 should be 4. Additionally as the most recent method is from 2011, I suggest not using the word "recent".

* Line 103: "AMINE can locate modules of 10 or 20 genes with a higher accuracy". Here you can just say "AMINE can locate modules with a higher accuracy", as "modules of 10 or 20" is confusing at this point. It is clear only after the rest of the manuscript is read.

**Evaluation of AMINE on artificial data generated by Robinson et al. (2017)

* Line 125: "standard deviation equals to ." The sentence is unfinished.

* Lines 121-122: "In this dataset, there are three modules to discover..." Are there three modules in each one of the 1000 graphs? This is not clear.

* In Supplemental_Fig_S1.png presents only the results for AMINE, can you put the results of MRF next to it? Precision, recall, F1 together are good enough to present the cases where AMINE finds small or large modules, so truncated recall seems unnecessary, also the random selection of nodes to have 30 nodes adds much uncertainty. I think this "truncated recall" metric can be removed. Another point, the y axis in panel A is "F1 score", it should be just "score" or "value".

* In Fig. 1 "GiGA" is written as "Giga"

**Validation of the method on artificial dense networks

- * Line 156: Instead of "several" you can give the exact number, because that number is of interest to the reader. I checked materials and methods for it, later I saw that this information was also available in the caption of the figure.
- * Lines 162-163: The repetition of "1000 vertices" and multiple "and"s complicate the sentence, please consider giving 1000 vertices information before this sentence, while/after telling network generation.
- * Lines 183-186: The modules are created at the sizes suitable for AMINE (Lines 478-480) and here it is said that AMINE finds modules with sizes close to the real ones'. This is not fair.
- * Lines 191-192: "This processing time is acceptable as real biological networks are close to this size." I do not understand the reasoning here. Please consider removing this sentence. The information that real biological networks have a size close to 10.000 can be given in or after the first sentence (Lines 188-189).
- * Lines 236-238: "Interestingly, three out of the five modules generated by AMINE were associated to ECM matrix organization (module 1 and 4) and cell interaction with ECM (module 3)." This sentence should be reshaped, "three out of five" does not fit with the rest of the sentence.

Discussion

- * Lines 334-336 "Despite the large number of methods developed over the last 20 years, AMINE identifies, with a higher accuracy than its competitors, modules created computationally on datasets intended to mimic the topology of biological networks."
This is a strong statement, the methods used for comparison are from the first half of that 20 years, and a subset of all tools. I suggest you reconsider this statement.

Methods

- * I think the information on Node2Vec parameters available in Supplemental Table S1 can be given in methods section instead of a supplementary file, if page limit allows.
- * Lines 450-453: This statement is not clear to me, how does expanding subnetwork as long as score keeps increasing overcomes the size bias, that as subnetwork size increases there is a higher chance to find a higher scoring subnetwork. Rather than overcoming, isn't your approach in line with the bias?
- * Lines 472-473 and Line 688: "3 initial nodes and setting parameters and to 0.09 and 0.7" The symbols for parameters are missing I guess.
- * It was stated that AMINE did not require any parameters, in the web site also I see that it does not ask for it, but in Algorithm 1 there is an input for the number of active modules to be identified. Can you explain / correct this?
- * In Algorithm 1, I guess C_i is sorted by scores, can you make this clear?
- * In Algorithm 1 the function "similar" is not clear, at first I thought it returned similarity measures but actually it returns the nodes most similar to v_i based on embeddings. Additionally, I guess "similar" uses a threshold to determine similar nodes, which is kind of parameter embedded in the code. Could you make these clear?
- * The powerset symbol in text and Algorithm 1 are hard to match.
- * On <http://amine.i3s.unice.fr/>, entering an email is currently optional. I did not try it but it seems like there is the possibility to retrieve the results just on web page, without the email, if you keep the browser open. Is it the case? Shouldn't the user be encouraged to enter an email with the explanation that run time may take an hour.

Reviewer #2 (Comments to the Authors (Required)):

In this manuscript, the authors propose Active Module Identification through Network Embedding (AMINE) as a method to detect modules in gene networks. The method is based on Node2vec for building clusters of nodes accordingly to the similarity -in terms of distance- of their encoding vectors mapped in a reduced dimensional space. Once the clusters are defined, their contained nodes are evaluated accordingly to their activity summarized as value per node (gene), generating an attributed gene network. In a first part, the authors described how they outperformed the method using an artificial dataset for identifying modules. Subsequently, they applied AMINE using a gene network from STRING and publicly available expression data from GEO of a transcriptomic data of pancreatic ductal adenocarcinoma cancer (PDAC) to test their method in real data. Finally, they performed in vitro experiments in order to attribute some apparent novel functions to Blimp1.

First of all, this method relies on combining interaction network and transcriptomic data to detect dataset-specific modules. As the authors state, there are other several methods using a similar approach, therefore it's essential to assess whether this manuscript provides a substantial advance. Unfortunately, this task is very complicated because several parts of this manuscript have been recently published by the same authors in this same journal, but analyzing instead expression data of Drosophila (DOI: 10.26508/lsa.202101119). I encounter this the biggest concern to recommend this article for publication.

In addition, I found other concerns in the study that are not fully supportive of clear advantages compared to previous methods and the authors should address additional analyses for contextualize better the performance of AMINE and its access as an app.

Comment 1.

The authors benchmark AMINE with other tools using an artificial data generated by Robinson et al. 2017. The authors took specific decisions in order to make a fair and apparently disadvantaged comparative for AMINE. However, the authors made the comparative using a fixed number (3 and 1) of hit modules and 1000 vertices.

Could the authors provide more analyses (or evidences) about the performance using a wider range of parameters? For instance, varying the number of hit modules and vertices, do the authors get a better overview of the AMINE performance?

Comment 2.

Embedding networks are a promising and accesible technique to summarize relevant relationships between nodes while retaining neighborhood data in a vector space. However, it's not clear the strength of node values in the method. Could the authors represent how is the relationship between similarity of encoding vectors of nodes and node proximity in the original network? Although the result could be even obvious, but this might be helpful for gaining an deeper overview of the effect of aggregated p-values.

Comment 3

Additionally, could the authors try other publicly available datasets of differentially expression studies? e.g. Expression Atlas is plenty of results that can be easily imported into multiple formats. These exploratory analysis would be beneficial to systematically asses the effect of aggregated p-values in architecture of the module as well as the specificity of the methods for independent datasets using the same original network.

Comment 4:

The authors focused mainly in protein-interaction network using STRING. I am missing in methods detailed and precise information about the used scores and cut-off used in order to reproduce the gene network. I saw that the authors included it in the already published paper. In line with this, the authors should compare their analysis building networks using source of scores from STRING in order to compare, for instance, the performance in noisy networks (i.e. based on text-mining) compared to experimentally validated interactions.

Comment 5:

Additionally, authors should discuss that AMINE has been performed in protein-interactions but there are other types of biological networks, such as metabolic or gene-regulatory networks, with different modular architectures that do not necessarily are incorporated STRING. These kind of biological networks are often disregarded in these methods and definitely would be step-forward.

Comment 6:

In my opinion, an actual step forward for AMINE would be to develop a package or at least to upload it in a repository such as GitHub.

Comment 7:

Blimp1 is an alias of Prdm1, and there are several literature relating Blimp1 to inflammatory processes. The authors should explain or specify a bit better in the manuscript what is the exact new function predicted for Blimp1.

MINOR COMMENTS (additional issues):

The figures based on STRING screenshots should be improved, in several points. Missing even a legend.

Footnote of Fig. 1 -> dense is an ambiguous term, maybe you should provide the number of edges.

L102 -> "AMINE outperforms 5 recent methods..." I can only account 3 different methods plus the expression value. If MRS is accounted, it would be preferable to add it in the Figures.

L125 -> " standard deviation equals to. " I think the SD value is missing.

L139 -> Typo at the semicolon, there is a blank space.

L400-L401 -> Typo double blank space?

L674 -> DeSeq2  DESeq2

Responses to the comments of the reviewers

Dear Reviewers,

We would like to thank you for taking the time to review our manuscript and for providing us with your valuable comments and corrections on our manuscript entitled "A network embedding approach to identify active modules in biological interaction networks". Your feedback has been extremely helpful in improving the quality of our work.

In response to your comments, we provide below a point-by-point response to address each of your concerns. For clarity, we have quoted your comments, and we have inserted our responses in blue font.

Reviewer #1 (Comments to the Authors (Required)):

In this study the authors suggest a network embedding based active module identification method. In this context modules are not constrained to be connected components but they are selected to be clusters in the embedding space. Network embedding is performed using biased random walk, in which authors defined their own bias: The transition probability between two nodes is high if their p-values are close to each other.

*** Main concerns ***

* The codes of AMINE should be shared on GitHub.

We have taken your suggestion into consideration and have decided to make our code available on GitHub. We have included the address of the GitHub repository in the article so that others can easily access and use the code.

* In Data Access section it is stated that "All data generated or analysed during this study are included in this published article and its supplementary information files."
I do not have access to these files, will they be available in the final publication?

You are correct in noting that our Data Access section lacks detailed information on how to access the data used in our study. We have since updated this section to provide more clarity on this matter. Specifically, we have made available the data from Robinson's article, which can be downloaded from the article itself, and has also been replicated on the AMINE GitHub repository. Additionally, we have placed the algorithm used to generate the synthetic data that we used to measure algorithm performance on GitHub. The RNA-Seq data from Chiou et al. (2017) can be downloaded from the GEO website and have also been duplicated on GitHub. The differential expression analyses performed on these data have also been made available on GitHub. We hope that these updates address your concerns and provide you with the information you need to access the data used in our study.

* AMINE was compared with a baseline method and three active module identification methods. The reason of choosing these tools for comparison is not justified. There are more recent methods,

such as DOMINO (<https://doi.org/10.15252/msb.20209593>). DOMINO was shown to be performing very well in its publication also in a review study (<https://doi.org/10.1093/bib/bbab066>). It is important to see the performance comparison of these methods, whether it is in favor of AMINE or not.

Thank you for your comment on our article. We compared our method with a baseline method and three active module identification methods that were selected based on the same dataset used by Robinson et al. in their paper published in 2017. In that paper, Robinson and colleagues compared their method with NEST, NePhe, Knode, and BioNet, which were proposed between 2009 and 2015, and demonstrated the superiority of their own method. We chose to compare our method with Robinson's method on the same artificial dataset that they generated and used to compare with the other methods. Supplementary Figure S1 shows the superiority of AMINE over Robinson's method and the four other methods.

We also compared our method with GiGA, BioNet, and COSINE, which were proposed in 2004, 2010, and 2011, respectively, on artificial datasets that we generated with more realistic and denser interaction networks. Overall, we compared our method, directly or indirectly, with many other methods proposed between 2004 and 2017.

However, you are right that there are more recent methods, such as DOMINO, which has shown promising results. Therefore, we included DOMINO and also DIAMOnD (a method proposed in 2015) in our tests and present and discuss the results in our manuscript. Thank you for bringing this to our attention, and we hope that our additional experiments with these newer methods will provide useful information for the scientific community.

* AMINE was compared with other methods only using artificial datasets. On real data, AMINE was evaluated by the true positive enrichment analysis results of the modules. The same can be done with the compared methods.

We agree that it is important to compare AMINE with other methods on real datasets to better understand the differences between them. In response to your comment, we conducted a comprehensive evaluation of AMINE and other methods on artificial and real datasets. This allowed us to compare the performance of AMINE with other methods on real data. We have updated our paper to include these results. We appreciate your feedback, as we feel it has helped us improve the quality of our paper.

* In the enrichment analysis results of the modules obtained using real data (Table 1) it is not clear whether the annotation terms are the most significant ones for those modules. How many other terms were enriched for each module? Could you comment on the precision of the enrichment results? I know that there is no ground truth, so I am not expecting a numerical value but similar to the way true positive enrichment results are commented on, a rough interpretation of precision can be done.

We recognize that we selected the annotations shown in Table 1 based on their importance and relevance to biologists, which introduces bias and does not provide a comprehensive view of the enrichments. To address this issue, as we now compare AMINE to other methods, we employed an objective way to select the most important annotation term for each module. Specifically, we only report the annotation with the lowest FDR among the terms in the KEGG, Reactome, WikiPathways, and Gene Ontology Biological Pathways databases. We mention this best annotation in the figures representing the modules (Figure 2 and Figures S7-S9) and have removed Table 1 and Table 2. In addition, to allow readers to access the full list of annotations, we have included them in supplemental Tables S1 and S5. We have also attempted to provide an

interpretation of the results obtained. Although there is no absolute truth, we believe that our approach provides a more objective and transparent way to report the results of the enrichment.

* AMINE does not constrain connectivity in the modules. To present how connected the nodes in each module are, can you plot edge density (number of edges / number of possible edges) for each module on a boxplot (as a supplementary file)?

We do not believe that edge density is a determining criterion in assessing the connectivity of modules. As an example, we can compare module 4, which contains only connected nodes and is identified by DOMINO (Fig. S7), with module 4, which contains an isolated node and is identified by AMINE (Fig. 2). Both modules contain 5 nodes, but the module with all connected nodes has fewer edges than the one with an isolated node. Additionally, it is important to consider that the edges in the networks are not ground truth, as the input networks are noisy and the topology of the modules can vary considerably depending on the cutoff values used. For instance, for the two modules mentioned above, the one identified by AMINE is a connected graph and has more edges than the one identified by DOMINO by just lowering the threshold to 0.6. Therefore, we believe that presenting the edge density in a boxplot would not be informative since it depends solely on how the input network is constructed.

* In active module identification, the selection of PPI network is obviously very important. Researchers usually favor a network over others based on how the interactions were identified or curated. In AMINE the user does not have an option to use another PPI. The user should have the option to upload their PPI or use the provided one.

In the current version of AMINE that we have deposited on GitHub, we provide the user with a choice of 3 sources for the PPI network: STRING, BioGRID and Intact. We also provide a file that represents the union of these three networks. In addition, users can upload their own PPI network and associated p-values using two simple files. We provide examples of these files so that users can easily adapt their own network for use with AMINE.

* A common approach while using STRING database is to filter interactions by confidence level (e.g. selecting the interactions with confidence score greater than 0.7) or source (e.g. excluding text-mined interactions). Did you apply any filtering?

Indeed, in the submitted version of our paper, we did not provide information on how the interactions are filtered. We thank the reviewer for pointing this out. We confirm that the version available on our website is based on the use of the PPI STRING network with interactions having a confidence score higher than 0.7, as it is usually done. In the GitHub repository, we provide three public PPI networks and the ability for users to upload their own network. The code also includes options for filtering PPI data. For example, for STRING, it is possible to filter the data based on each component of the confidence score. By default, we use a filter that only includes interactions with an overall confidence score greater than 0.7. For BioGRID, users can filter interactions based on the type of interaction (physical or genetic). By default, all interactions are used. For IntAct, it is possible to filter based on a minimum value of the confidence value. By default, all interactions are retrieved. We have added all this information in the material section.

* The bias in RWR that is introduced by the authors (defined in Methods - Network embedding section) should be discussed. In active module identification, if a node is connected to significant nodes, then there is a chance that it is involved in the module ("guilt by association"). However in the approach followed here, nodes with similar level of significance will be more similar in the

vector representation, thus they will be clustered, which will decrease the chance of an insignificant node being involved in the module.

In our method, nodes with similar levels of significance are indeed more likely to be similar in the vector representation and thus to be grouped together, which can reduce the probability of an insignificant node being involved in the module. However, our method is not completely free of the network topology since, during the random walk, the transition from one node to another can only occur if a link exists. The probability of moving from a highly significant node to a non-significant node is never equals to zero.

In the bias formula we introduced in the Methods - Network Integration section, the only way to have a transition probability of zero between two nodes is for node i to be associated with a value of 1 and node j with a value of 0 (or vice versa). In our program, we assign a value of epsilon to this case, to ensure that all transitions are possible. However, we recognize that we did not specify this in the text of the article. We have since modified our manuscript to express the bias using the formula $t_{i,j} = \max(1 - |p_i - p_j|, e)$ with e being a very small value (concretely set to $1e-16$ in the algorithm) that prevents obtaining a transition probability between two nodes equal to zero.

* On <http://amine.i3s.unice.fr/>, there is no initial checking of the submitted files (I tried some files). If there is a problem in the input file the user should be notified early.

Thank you for bringing up this important point. We agree with the reviewer that it is crucial to perform an initial check of the files submitted to our website to ensure that any potential problems are flagged early on. However, we would like to clarify that the website was primarily designed to allow our fellow biologists to use our tool. In light of the reviewer's comment, we decided that it is not appropriate to make the application widely accessible, as our hosting capacity is limited, both in terms of CPU and bandwidth. Instead, we prioritized distributing the program's sources on GitHub, so that anyone can run it freely. We now emphasize this option in the manuscript. Nevertheless, we keep the website available as it exists, and it may be useful for people without computer science skills to use the application. However, this access should only be considered as a site for quick testing of the method or occasional analysis. We included a statement in the manuscript to clarify these new provisions.

* On <http://amine.i3s.unice.fr/>, with each consecutive submission (submitted before the previous one ends) waiting time increases one hour, seems like one job can be run at a time. In this state, making people use this tool might be hard.

This issue is related to the limited hosting capacity we have, as mentioned in our response to the previous comment. We recognize that this can make it difficult to use the tool via the website, and we would like to emphasize that website access should now only be considered as an alternative to local installation, which we strongly encourage. We included a statement in the manuscript to make this clear to the readers.

*** Minor concerns ***

**Introduction

* In this study the literature on active module identification is very well presented. However there is not a mention of any other study that uses network embedding to prioritize genes/proteins (e.g. <https://doi.org/10.3389/fgene.2019.00226>). The literature on this subject should also be mentioned.

Thank you for your feedback. We have taken it into account and updated the manuscript accordingly. We have added in the introduction a paragraph discussing various computational biology studies that utilize network embedding for prioritizing genes/proteins, including the study you mentioned (<https://doi.org/10.3389/fgene.2019.00226>). We appreciate your input and are glad to have improved the manuscript with your help.

* "Active module" generally refers to a set of connected nodes, here there is no constraint for connectedness. This is not mentioned until the Methods section, please make your definition of module earlier (in Introduction).

Thank you for your suggestion. We have revised the manuscript accordingly and added a clarification in the Introduction. Specifically, we now state that "[...] the identified modules may not necessarily be fully connected in the original graph structure, in contrast to other methods that operate on the graph. Our approach thus favors the proximity of the nodes in the reduced vector space but there is no constraint for connectedness." This should clarify our approach and the definition of modules in our study.

* Lines 38-40: The sentence "In order to ..." is hard to read/understand at first sight, it might be improved.

We have replaced this sentence with: «To identify the most relevant genes for the process under study, statistical methods are commonly used to assign numerical values to the genes which reflect the degree of variation observed.»

* Lines 53-54: "These pertinent subnetwork are" -> "subnetworks"

We have made the correction in the manuscript.

* Lines 54-55: Some of the names are plural, some are singular, it will be better if they are consistent.

We have made the correction in the manuscript.

* Line 57: "one based on a measurement of genes activity" -> "one based on the measurement of genes' activities"

We have made the correction in the manuscript.

* Line 69: It is not clear what it is meant with distance (distance on network or opposite of similarity), also it is not clear why relying on distance is bad.

Thank you for bringing this to our attention. You are right that the sentence in question is unclear. What we meant was that clustering algorithms rely on a measure of distance or similarity between objects, which can be difficult to define on an attributed graph. This distance measure must combine both a topological distance (e.g., the number of edges separating two nodes) and the similarity of values associated with the nodes. The challenge is to determine an appropriate way to compute this distance measure. We have revised the sentence as follows: "Finally, methods (vi) offers the advantage of being based on existing clustering algorithms, but they require the calculation of a distance (or similarity) metric between objects. On an attributed graph, this distance must combine topological distances (such as the number of edges separating two nodes) with the similarity of the values associated with the nodes, making it challenging to determine the appropriate metric."

* Lines 73-88: There is a problem with the story flow here, at line 80 there is Node2Vec, but then deep learning and word embedding came... Shouldn't word embedding be mentioned before Node2Vec, if necessary?

You are right. We have moved the sentence introducing Node2vec to the end of the following paragraph.

* Lines 97-99: "a greedy approach to build increasingly large clusters of nodes". As a reader who knows how greedy approach works, this sentence made me think like there is an extra process to the greedy approach, like several clusters with different sizes were built, however it is just the greedy approach expanding the cluster. I suggest you to reconsider this sentence. Another point, the sentence is like the greedy approach is used to evaluate the clusters. Could it be like "Our method uses a greedy approach to build the clusters based on the similarity of the nodes' encoding vectors and a metric that takes into account the activity of the contained nodes." ?

Thank you for your valuable suggestion. We agree that the sentence was unclear and could have been misleading. We have considered your suggestion and have included the replacement sentence you suggest in our revised manuscript.

* Line 102: "AMINE outperforms 5 recent methods". 5 should be 4. Additionally as the most recent method is from 2011, I suggest not using the word "recent".

The reference to "5 recent methods" in line 102 was intended to include the MRF method and four other methods to which it was compared. However, we understand that the wording may have been unclear. We have revised our sentence as follows (deleting the word "recent"): "On sparse interaction networks, in a task consisting in finding 3 distinct gene modules, AMINE outperforms the MRF method (Robinson et al. 2017), which itself achieved better results than four other methods published between 2009 and 2015, using the exact same dataset."

* Line 103: "AMINE can locate modules of 10 or 20 genes with a higher accuracy". Here you can just say "AMINE can locate modules with a higher accuracy", as "modules of 10 or 20" is confusing at this point. It is clear only after the rest of the manuscript is read.

We have taken your comment into consideration and made the change accordingly.

**Evaluation of AMINE on artificial data generated by Robinson et al. (2017)

* Line 125: "standard deviation equals to ." The sentence is unfinished.

The standard deviation used is equal to 0.05 as in the article by Robinson et al. We have corrected the sentence.

* Lines 121-122: "In this dataset, there are three modules to discover..." Are there three modules in each one of the 1000 graphs? This is not clear.

To clarify, there are indeed three distinct modules to be identified in each of the 1000 graphs in the dataset. We have modified the original sentence as follows: "In this dataset, each graph contains three distinct modules to be identified (called "hit modules"), with each module containing 10 vertices" in order to eliminate any ambiguity regarding the number of modules per graph. We thank you for your input and hope this clarification is helpful."

* In Supplemental_Fig_S1.png presents only the results for AMINE, can you put the results of MRF next to it?

Precision, recall, F1 together are good enough to present the cases where AMINE finds small or large modules, so truncated recall seems unnecessary, also the random selection of nodes to have 30 nodes adds much uncertainty. I think this "truncated recall" metric can be removed.

Another point, the y axis in panel A is "F1 score", it should be just "score" or "value".

We purposely did not include the MRF results in Fig. S1 because Robinson et al. do not provide precise numerical results for their method and only present their results in a figure. The boundaries in their figure are imprecise, and we did not want to distort Robinson et al.'s data. However, we have added a comparison of the results in the main text where we can be more nuanced. We followed the reviewer's recommendation to remove the truncated recall metric both from the text and the figure. Additionally, we modified the x-axis label in the figure.

* In Fig. 1 "GiGA" is written as "Giga"

We have made the correction in the manuscript.

**Validation of the method on artificial dense networks

* Line 156: Instead of "several" you can give the exact number, because that number is of interest to the reader. I checked materials and methods for it, later I saw that this information was also available in the caption of the figure.

We have specified this information in the manuscript.

* Lines 162-163: The repetition of "1000 vertices" and multiple "and"s complicate the sentence, please consider giving 1000 vertices information before this sentence, while/after telling network generation.

We have simplified this sentence.

* Lines 183-186: The modules are created at the sizes suitable for AMINE (Lines 478-480) and here it is said that AMINE finds modules with sizes close to the real ones'. This is not fair.

From a biological perspective, we focused on discovering small modules as identifying modules of several hundred genes does not provide much insight. Therefore, we created datasets where small modules had to be identified. AMINE performs well in finding modules of varying sizes without any prior knowledge of the true sizes. This behavior contrasts with many methods that require the module sizes to be specified as a parameter. We challenge the fact that we have biased the dataset to give our method an unfair advantage.

* Lines 191-192: "This processing time is acceptable as real biological networks are close to this size." I do not understand the reasoning here. Please consider removing this sentence. The information that real biological networks have a size close to 10.000 can be given in or after the first sentence (Lines 188-189).

We agree, this sentence is not necessary. We have deleted it.

* Lines 236-238: "Interestingly, three out of the five modules generated by AMINE were associated to ECM matrix organization (module 1 and 4) and cell interaction with ECM (module 3)." This sentence should be reshaped, "three out of five" does not fit with the rest of the sentence.

We have rewritten the sentence as follows : Notably, three out of the five modules produced by AMINE were found to be linked to the stromal reaction (desmoplasia): modules 1 and 4 were enriched for “organization of the extracellular matrix (ECM)” and module 3 for “cellular interactions with the ECM”

Discussion

* Lines 334-336 "Despite the large number of methods developed over the last 20 years, AMINE identifies, with a higher accuracy than its competitors, modules created computationally on datasets intended to mimic the topology of biological networks."

This is a strong statement, the methods used for comparison are from the first half of that 20 years, and a subset of all tools. I suggest you reconsider this statement.

We have taken your feedback into consideration and have revised our statement accordingly. We would also like to inform you that we are now including a comparison of our method with DOMINO, a newly published method in 2021, in addition to other existing tools. Our updated statement reads as follows: "Although many methods have been developed over the past two decades, AMINE stands out for its ability to accurately identify modules on datasets designed to mimic the structure of biological networks, outperforming many other competing methods".

Methods

* I think the information on Node2Vec parameters available in Supplemental Table S1 can be given in methods section instead of a supplementary file, if page limit allows.

We have done this. Thank you for your suggestion.

* Lines 450-453: This statement is not clear to me, how does expanding subnetwork as long as score keeps increasing overcomes the size bias, that as subnetwork size increases there is a higher chance to find a higher scoring subnetwork. Rather than overcoming, isn't your approach in line with the bias?

We agree that our statement may not be intuitive and can be misleading. Upon further investigation, we have found that large random networks can indeed have higher scores than smaller networks consisting of genes of interest. However, it is generally difficult (if not impossible) to obtain such large networks through the incremental addition of nodes while maintaining a monotonic increase in the score. Given the lack of evidence to support this claim, we have removed the reference to this bias and Nikolayeva's article from our paper. Thank you for bringing this to our attention.

* Lines 472-473 and Line 688: "3 initial nodes and setting parameters and to 0.09 and 0.7" The symbols for parameters are missing I guess.

Thank you for your feedback. The reviewer is correct, and we apologize for the oversight. The values mentioned refer to the parameters p and q used in Barabasi's article. We have made the necessary changes to the text accordingly. We appreciate the reviewer's keen attention to detail.

* It was stated that AMINE did not require any parameters, in the web site also I see that it does not ask for it, but in Algorithm 1 there is an input for the number of active modules to be identified. Can you explain / correct this?

It is correct that the AMINE method does not require the number of active modules to be identified as an input parameter. The parameter "n" that appears in Algorithm 1 is solely used to filter the results by returning only the top "n" identified modules. Therefore, there is no reason for it to appear in the method. We have corrected the algorithm accordingly.

* In Algorithm 1, I guess C_i is sorted by scores, can you make this clear?

You are correct that the nodes in C_i are sorted by their scores in descending order. We have modified the algorithm to make it clearer, in particular by no longer using the function "similar" whose name is inappropriate, as you pointed out in your following comment

* In Algorithm 1 the function "similar" is not clear, at first I thought it returned similarity measures but actually it returns the nodes most similar to v_i based on embeddings. Additionally, I guess "similar" uses a threshold to determine similar nodes, which is kind of parameter embedded in the code. Could you make these clear?

You are correct that the name of the "similar" function is poorly chosen and brings a lot of confusion. We agree that the function name suggests that it returns similarity measures, whereas it actually returns the nodes most similar to v_i . Furthermore, we understand that the function name "similar" also suggests that there is a selection of the closest nodes using a threshold. This is not the case. In the method, a sorting of all the nodes in the embedding is performed based on their cosine distance from the node under consideration. To address these issues, we have modified the algorithm and the manuscript accordingly, explicitly using a function named "sort" to make it clearer.

* The powerset symbol in text and Algorithm 1 are hard to match.

We have discovered errors in the algorithm as the version presented in the manuscript corresponds to a previous implementation. We apologize for this mistake. In the current version, the powerset is no longer used. We have now corrected the algorithm and verified that the pseudocode matches the Python implementation deposited on GitHub.

* On <http://amine.i3s.unice.fr/>, entering an email is currently optional. I did not try it but it seems like there is the possibility to retrieve the results just on web page, without the email, if you keep the browser open. Is it the case? Shouldn't the user be encouraged to enter an email with the explanation that run time may take an hour.

Yes, that is correct. Currently, entering an email address is optional, as it is possible to retrieve results from the web page, as long as the user keeps the browser open. In response to your suggestion, we have updated the website. This update makes it clear that processing time can take several hours, depending on the jobs in queue, and we now recommend that users enter their email address so that they can be notified when their results are ready.

Reviewer #2 (Comments to the Authors (Required)):

In this manuscript, the authors propose Active Module Identification through Network Embedding (AMINE) as a method to detect modules in gene networks. The method is based on Node2vec for building clusters of nodes accordingly to the similarity -in terms of distance- of their encoding vectors mapped in a reduced dimensional space. Once the cluster are defined, their contained nodes

are evaluated accordingly to their activity summarized as value per node (gene), generating an attributed gene network. In a first part, the authors described how they outperformed the method using an artificial dataset for identifying modules. Subsequently, they applied AMINE using a gene network from STRING and publicly available expression data from GEO of a transcriptomic data of pancreatic ductal adenocarcinoma cancer (PDAC) to test their method in real data. Finally, they performed in vitro experiments in order to attribute some apparent novel functions to Blimp1.

First of all, this method relies on combining interaction network and transcriptomic data to detect dataset-specific modules. As the authors state, there are other several methods using a similar approach, therefore it's essential to assess whether this manuscript provides a substantial advance. Unfortunately, this task is very complicated because several parts of this manuscript have been recently published by the same authors in this same journal, but analyzing instead expression data of Drosophila (DOI: 10.26508/lsa.202101119). I encounter this the biggest concern to recommend this article for publication.

We understand your concern regarding the recent publication of similar work by our group in this journal. However, we would like to emphasize that this manuscript provides a substantial advance in our research as it presents a detailed description of the Active Module Identification through Network Embedding (AMINE) method. In contrast, our previous publication focused on the application of the method to expression data from Drosophila.

We acknowledge that the publication timeline of our work may have caused some confusion. To clarify, the AMINE method was developed by our group and has been used by multiple collaborating teams. One of these collaborations resulted in interesting results, which we submitted for publication citing the bioArXiv reference of the present article. The application article was accepted before the method article, leading to a chronological issue. We understand that this may have caused confusion, but we would like to assure you that this manuscript provides a complete and detailed description of the AMINE method, which has not been previously published.

Furthermore, we would like to highlight the improvements we have made to the method, which we believe make it even more valuable. We have conducted a detailed analysis of the method's performance on a real gene network and expression data from pancreatic ductal adenocarcinoma cancer (PDAC). Specifically, we compared AMINE's capabilities with other methods on this dataset. Additionally, we have made all the sources publicly available on a repository, considered several different PPI networks, and provided the option for users to upload their own network to use with AMINE.

In addition, I found other concerns in the study that are not fully supportive of clear advantages compared to previous methods and the authors should address additional analyses for contextualize better the performance of AMINE and its access as an app.

In the revised version of our manuscript, we addressed your concern by comparing AMINE with other methods on artificial and real-world data sets. We reanalyzed Chiou et al.'s data using GiGA, BioNet, DIAMOND, and DOMINO, and provided an interpretation of the results obtained. Although in the case of real data, there is no ground truth, we highlighted the pros and cons of each method and pointed out the strengths of AMINE over the other methods. We hope that the new evidence we provide will help to better understand the capabilities of AMINE relative to competing methods. Our goal was provide a comprehensive and detailed analysis of AMINE's performance and to contextualize its results against other leading methods. In doing so, we believe our work will help researchers in the field to make informed decisions about which methods to use for their specific research questions and datasets.

Comment 1.

The authors benchmark AMINE with other tools using an artificial data generated by Robinson et al. 2017. The authors took specific decisions in order to make a fair and apparently disadvantaged comparative for AMINE. However, the authors made the comparative using a fixed number (3 and 1) of hit modules and 1000 vertices.

Could the authors provide more analyses (or evidences) about the performance using a wider range of parameters? For instance, varying the number of hit modules and vertices, do the authors get a better overview of the AMINE performance?

The Robinson et al. dataset consists of 3 modules of 10 nodes on a network of 1000 nodes, and we used exactly the same data that cannot be modified. We also tested the performance of AMINE on denser artificial networks of 1000 and 10,000 nodes, and looked for modules of 10 and 20 nodes. This represents several configurations (1 module vs. 3 modules, module size of 10 vs. 20, network size of 1000 vs. 10,000) and considerable computational time. We questioned whether more was needed, and given that the ultimate goal of the method is to detect small modules, we felt that our tests provided sufficient insight into the method's behavior. It is important to keep in mind that the goal is to identify active modules in real data. Artificial networks are only an approximation of real data and their relevance cannot be measured. This is why we believe that tests on artificial networks are only performed to get an idea of the algorithm's behavior and should not be the sole focus. Testing on real data is more informative in this sense, even if we do not have an objective measure to quantify the performance of the method.

Comment 2.

Embedding networks are a promising and accessible technique to summarize relevant relationships between nodes while retaining neighborhood data in a vector space. However, it's not clear the strength of node values in the method. Could the authors represent how is the relationship between similarity of encoding vectors of nodes and node proximity in the original network? Although the result could be even obvious, but this might be helpful for gaining an deeper overview of the effect of aggregated p-values.

We agree that illustrating the relationship between the similarity of node encoding vectors and the proximity of nodes in the original network can help provide a more thorough overview of the effect of our proposed biased random walk. To this end, we based our analysis on a toy example to clearly visualize the effect of bias. We presented the toy graph and the three-dimensional embeddings obtained using standard Node2Vec and the biased random walk version. We believe this will provide valuable insight into the impact of using our biased random walk for embedding.

Comment 3

Additionally, could the authors try other publicly available datasets of differentially expression studies? e.g. Expression Atlas is plenty of results that can be easily imported into multiple formats. These exploratory analysis would be beneficial to systematically asses the effect of aggregated p-values in architecture of the module as well as the specificity of the methods for independent datasets using the same original network.

We agree that such exploratory analyses could be beneficial for systematically evaluating our method on independent data sets. However, we would like to point out that performing such analyses would require a great deal of time and effort, not only in extracting and processing the data with AMINE, but especially in interpreting the results. In addition, adding all these analyses to the manuscript would make it very large and might distract from the main objective, namely

the description of the proposed method. It should be noted that several published papers already include analyses performed with AMINE. In addition to the study mentioned by the reviewer, we also mention the following two articles, which provide an overview of what is possible with our method :

- *Feliz Morel AJ et al. (2022). Persistent Properties of a Subpopulation of Cancer Cells Overexpressing the Hedgehog Receptor Patched. *Pharmaceutics*, 14(5), 988.*
- *Pasquier C & Robichon A (2022). Evolutionary Divergence of Phosphorylation to Regulate Interactive Protein Networks in Lower and Higher Species. *International Journal of Molecular Sciences*, 23(22), 14429.*

We have added a few lines to our manuscript to refer to these previous analyses performed with AMINE.

Comment 4:

The authors focused mainly in protein-interaction network using STRING. I am missing in methods detailed and precise information about the used scores and cut-off used in order to reproduce the gene network. I saw that the authors included it in the already published paper. In line with this, the authors should compare their analysis building networks using source of scores from STRING in order to compare, for instance, the performance in noisy networks (i.e. based on text-mining) compared to experimentally validated interactions.

Our article, in its submitted form, did not include details on the filtering of interactions. We are grateful to the reviewer for bringing this to our attention. As a clarification, we would like to confirm that the version of our method used to perform the experiment described in the manuscript utilizes the PPI STRING network with interactions that have a confidence score above 0.7, which is a commonly used threshold. The source code on GitHub includes options to filter PPI data according to different criteria. For instance, for STRING, users can filter interactions based on individual components of the confidence score. It is therefore quite easy to use AMINE on networks containing only validated interactions or only those obtained by text mining.

With regards to the second part of the comment, we would like to highlight, as we did above, that the workload is not in the construction of the networks nor in the execution of AMINE but in the analysis of the results. However, the reviewer's suggestion to compare the results obtained on the same dataset using different networks is very interesting and we thank him for this idea. In response to this comment, we have therefore included a subsection in the methods section of our manuscript entitled "Evaluating the algorithm's resistance to noisy data." In this section, we analyzed the Chiou et al. data using different interaction networks, including BioGRID, Intact, STRING with a global score threshold set to 0.9 and 0.4 (in addition to the threshold of 0.7 in the article) and STRING applying a threshold of 0.7 on the different components of the global score. We show that the AMINE method performs well regardless of the network, even with the one obtained from text mining which can be considered as noisy and that the default use of the STRING database with a threshold of 0.7 on the global score seems relevant, as it allows to obtain enrichment results similar to those obtained by using a threshold of 0.4 with a smaller dataset, thus reducing the execution time of the algorithm.

Comment 5:

Additionally, authors should discuss that AMINE has been performed in protein-interactions but there are other types of biological networks, such as metabolic or gene-regulatory networks, with different modular architectures that do not necessarily are incorporated STRING. These kind of biological networks are often disregarded in these methods and definitely would be step-forward.

We appreciate your suggestion that AMINE should not be limited to protein-protein interaction networks, and we agree that other types of biological networks, such as metabolic or gene-regulatory networks, could be valuable to incorporate into the method. In our article, we focused on using PPI network from STRING but additionally, we have provided the option for users to upload their own network in a simple format, which would enable the use of AMINE with other types of networks. We agree that incorporating other types of networks would be a valuable step forward. We mentioned these perspectives on the use of AMINE at the end of the discussion.

Comment 6:

In my opinion, an actual step forward for AMINE would be to develop a package or at least to upload it in a repository such as GitHub.

We agree that having a package or repository for AMINE would be a significant improvement. In fact, the first reviewer also made a similar suggestion. As a result, we have deposited the source code of the method on GitHub. This will make it easier for other researchers to use and evaluate AMINE in their own work.

Comment 7:

Blimp1 is an alias of Prdm1, and there are several literature relating Blimp1 to inflammatory processes. The authors should explain or specify a bit better in the manuscript what is the exact new function predicted for Blimp1.

We agree that BLIMP1/PRDM1 is already known as a master regulator of hematopoietic stem cells, and plays a critical role in the development of plasma B cells, T cells, dendritic cells (DCs), macrophages, and osteoclasts. BLIMP-1 is also a gatekeeper of T-cell activation and plays a key role in maintaining normal T cell homeostasis. In their paper, Chiou et al. (2017) demonstrated that Blimp1 controls hypoxia-activated EMT process in epithelial cancer cells. In our study, we demonstrate using AMINE that Blimp1 also activates pro-inflammatory response in PDAC epithelial cell, which has never been found before. This information have been included in the new version of the manuscript.

MINOR COMMENTS (additional issues):

The figures based on STRING screenshots should be improved, in several points. Missing even a legend.

We have taken note of your comment regarding the figures based on STRING screenshots and have updated them accordingly.

Footnote of Fig. 1 -> dense is an ambiguous term, maybe you should provide the number of edges.

We agree that the term dense is not precise enough. However, we would like to clarify that since the networks were randomly generated, the number of edges differs for each network. We have chosen to specify in the legend of the figure the average number of edges which is around 4300.

L102 -> "AMINE outperforms 5 recent methods..." I can only account 3 different methods plus the expression value. If MRS is accounted, It would be preferable to add it in the Figures.

The mention of "5 recent methods" was meant to encompass the MRF method and four other techniques it was compared against. We recognize that the phrasing may have been unclear, and thus we have amended our statement as follows: "On sparse interaction networks, in a task consisting in finding 3 distinct gene modules, AMINE outperforms the MRF method (Robinson

et al. 2017), which itself achieved better results than four other methods published between 2009 and 2015, using the exact same dataset."

L125 -> " standard deviation equals to. " I think the SD value is missing.

The standard deviation used is equal to 0.05 as in the article by Robinson et al. We have corrected the sentence.

L139 -> Typo at the semicolon, there is a blank space.

L400-L401 -> Typo double blank space?

L674 -> DeSeq2  DESeq2

These typos have been corrected.

May 25, 2023

RE: Life Science Alliance Manuscript #LSA-2022-01550-TR

Dr. Claude Pasquier
French National Centre for Scientific Research
Laboratoire I3S
CS 40121
Sophia Antipolis 06903
France

Dear Dr. Pasquier,

Thank you for submitting your revised manuscript entitled "A network embedding approach to identify active modules in biological interaction networks". We would be happy to publish your paper in Life Science Alliance pending final revisions necessary to meet our formatting guidelines.

- please address the final points of both Reviewers
- please upload your Table S7 in editable .doc or excel format
- please consult our manuscript preparation guidelines <https://www.life-science-alliance.org/manuscript-prep> and make sure your manuscript sections are in the correct order
- please add a Summary Blurb in your main manuscript
- please rename your Data Access section with Data Availability
- please use the [10 author names, et al.] format in your references (i.e. limit the author names to the first 10)

Figure check:

- Figure 4A,C: Please indicate molecular weight next to each protein blot
- please provide a figure legend for Figure 7
- please provide higher resolution image for Supplementary Figure 2,3,10. Minimum resolution for all figures is 300 dpi. For figures that contain both photographs and line art or text, 600 dpi is highly recommended.
- please add a callout for Figure 1A, Figure S1A, Figure S5B, Figure S6A and B, Table S2, S8, S9, S10, S11, S12, S13, S14, S15 to your main manuscript text

A. FINAL FILES:

- An editable version of the final text (.DOC or .DOCX) is needed for copyediting (no PDFs).
- High-resolution figure, supplementary figure and video files uploaded as individual files: See our detailed guidelines for preparing your production-ready images, <https://www.life-science-alliance.org/authors>
- Summary blurb (enter in submission system): A short text summarizing in a single sentence the study (max. 200 characters)

including spaces). This text is used in conjunction with the titles of papers, hence should be informative and complementary to the title. It should describe the context and significance of the findings for a general readership; it should be written in the present tense and refer to the work in the third person. Author names should not be mentioned.

B. MANUSCRIPT ORGANIZATION AND FORMATTING:

Sincerely,

Reviewer #1 (Comments to the Authors (Required)):

I thank the authors for the revision. I have a few comments / suggestions.

Line 152-163: In this part, you say that Barabasi-Albert model of preferential attachment is too far from a real interaction graph. Then there is the sharp transition to using the extended version of this model. Was there an enormous development from the initial model to the extended version? As I read further I see that you spent effort to find good parameters to create realistic networks, did this convince you to use this model? This part is confusing for me. Please reshape / improve the writing to make these two paragraphs more consistent. Additionally, how do you evaluate these networks' similarity to real networks?

Line 380-383: For the incompleteness of the PPI it is better to refer to a more recent article. Some examples: PMID: 36009835, PMID: 28284537, PMID: 25700523

In Algorithm 1, "score" should be reassigned in the while loop (after line 9).

Line 92: variery -> variety

Line 97: active modules identification -> active module identification

Line 115: 2015. using -> 2015 using

Line 119: pathways/biological process -> pathways/biological processes

Line 238: EMT acronym was used before its full form (line 259)

Line 244: ECM acronym was used before its full form (line 297)

Line 384: "We think first of all of the methods based on the identification of cliques" -> Can you improve this sentence?

Reviewer #2 (Comments to the Authors (Required)):

In this revised manuscript, the authors have addressed the two main concerns: (i) providing more details of the specific novelty of this work compare to previous applications and (ii) making available AMINE in GitHub which is a great advance for AMINE.

The authors have mostly addressed other comments and this new manuscript has improved considerably.

As a final new minor comment, I suggest the authors to contextualise these recent update in node2vec+, <https://doi.org/10.1093/bioinformatics/btad047>

Responses to the comments of the reviewers

Dear Reviewers,

We greatly appreciate your efforts in reviewing our manuscript titled "A network embedding approach to identify active modules in biological interaction networks" and providing us with your valuable comments and corrections. Your feedback has played a crucial role in enhancing the overall quality of our work, and we are sincerely grateful for your time and input.

In response to your comments, we provide below a point-by-point response to address each of your concerns. For clarity, we have quoted your comments, and we have inserted our responses in blue font.

Scientific Editor

Thank you for submitting your revised manuscript entitled "A network embedding approach to identify active modules in biological interaction networks". We would be happy to publish your paper in Life Science Alliance pending final revisions necessary to meet our formatting guidelines.

- please address the final points of both Reviewers
- please upload your Table S7 in editable .doc or excel format
- please consult our manuscript preparation guidelines <https://www.life-science-alliance.org/manuscript-prep> and make sure your manuscript sections are in the correct order
- please add a Summary Blurb in your main manuscript
- please rename your Data Access section with Data Availability
- please use the [10 author names, et al.] format in your references (i.e. limit the author names to the first 10)

All these points were checked and corrected.

Figure check:

- Figure 4A,C: Please indicate molecular weight next to each protein blot
- please provide a figure legend for Figure 7
- please provide higher resolution image for Supplementary Figure 2,3,10. Minimum resolution for all figures is 300 dpi. For figures that contain both photographs and line art or text, 600 dpi is highly recommended.
- please add a callout for Figure 1A, Figure S1A, Figure S5B, Figure S6A and B, Table S2, S8, S9, S10, S11, S12, S13, S14, S15 to your main manuscript text

The requested corrections have been made in the new version of the manuscript. Regarding the last point : tables S8 to S15 are referred to in the manuscript with "Tables S8-S15".

Reviewer #1 (Comments to the Authors (Required)):

I thank the authors for the revision. I have a few comments / suggestions.

Line 152-163: In this part, you say that Barabasi-Albert model of preferential attachment is too far from a real interaction graph. Then there is the sharp transition to using the extended version of this model. Was there an enormous development from the initial model to the extended version? As I read further I see that you spent effort to find good parameters to create realistic networks, did this convince you to use this model? This part is confusing for me. Please reshape / improve the writing to make these two paragraphs more consistent. Additionally, how do you evaluate these networks' similarity to real networks?

We acknowledge that this part of the manuscript is unclear. We have rewritten it by explaining the preferential attachment model and its evolution proposed in 2000 by Barabási and Albert. There has been a significant evolution of the model proposed in 2020, which uses two additional parameters (p and q) to generate more realistic networks. As you have noticed, we have put a lot of effort into finding the parameter values that enable to generate networks topologically close to real PPI networks. We evaluated the similarity of the generated networks with real networks by comparing several metrics, including the alpha coefficient, the R square and the mean of neighbor connectivity.

Line 380-383: For the incompleteness of the PPI it is better to refer to a more recent article. Some examples: PMID: 36009835, PMID: 28284537, PMID: 25700523

We have taken your remark into consideration and included the reference to “Kondratyeva et al. 2022”.

In Algorithm 1, "score" should be reassigned in the while loop (after line 9).

You are completely right. It is indeed an error in the algorithm. Thank you very much for bringing this to our attention. We have made the necessary correction.

Line 92: variery -> variety

Line 97: active modules identification -> active module identification

Line 115: 2015. using -> 2015 using

Line 119: pathways/biological process -> pathways/biological processes

Line 238: EMT acronym was used before its full form (line 259)

Line 244: ECM acronym was used before its full form (line 297)

All these points were corrected.

Line 384: "We think first of all of the methods based on the identification of cliques" -> Can you improve this sentence?

We have rewritten this passage as follows: “This observation suggests that methods relying heavily on network topology may not be very suitable. In particular, the effectiveness of methods based on clique identification may be questionable if we consider that a significant proportion of protein-protein interactions remain unknown”.

Reviewer #2 (Comments to the Authors (Required)):

In this revised manuscript, the authors have addressed the two main concerns: (i) providing more details of the specific novelty of this work compare to previous applications and (ii) making available AMINE in GitHub which is a great advance for AMINE.

The authors have mostly addressed other comments and this new manuscript has improved considerably.

As a final new minor comment, I suggest the authors to contextualise these recent update in node2vec+, <https://doi.org/10.1093/bioinformatics/btad047>

Thank you for your valuable comment. In our manuscript, we have taken into account the recent advances in the field and have provided a contextualization of our approach in light of these updates.

June 6, 2023

RE: Life Science Alliance Manuscript #LSA-2022-01550-TRR

Dr. Claude Pasquier
French National Centre for Scientific Research
Laboratoire I3S
CS 40121
Sophia Antipolis 06903
France

Dear Dr. Pasquier,

Thank you for submitting your Methods entitled "A network embedding approach to identify active modules in biological interaction networks". It is a pleasure to let you know that your manuscript is now accepted for publication in Life Science Alliance. Congratulations on this interesting work.

DISTRIBUTION OF MATERIALS:

Again, congratulations on a very nice paper. I hope you found the review process to be constructive and are pleased with how the manuscript was handled editorially. We look forward to future exciting submissions from your lab.

Sincerely,
